# Enhancing Large Vision Language Models with Self-Training on Image Comprehension

**Yihe Deng**[*1], **Pan Lu**[*1,3], **Fan Yin**[1], **Ziniu Hu**[1], **Sheng Shen**[2]
**Quanquan Gu**[1], **James Zou**[3], **Kai-Wei Chang**[1], **Wei Wang**[1]

[1]University of California, Los Angeles
[2]University of California, Berkeley    [3]Stanford University

https://stic-lvlm.github.io/

## Abstract

Large vision language models (LVLMs) integrate large language models (LLMs) with pre-trained vision encoders, thereby activating the model's perception capability to understand image inputs and conduct subsequent reasoning for different queries. Improving this capability requires high-quality vision-language data, which is costly and labor-intensive to acquire. Self-training approaches have been effective in single-modal settings to alleviate the need for labeled data by leveraging model's own generation. However, effective self-training remains a challenge regarding the unique visual perception and reasoning capability of LVLMs. To address this, we introduce **S**elf-**T**raining on **I**mage **C**omprehension (**STIC**), which emphasizes a self-training approach specifically for image comprehension. First, the model self-constructs a preference dataset for image descriptions using unlabeled images. Preferred responses are generated through a step-by-step prompt, while dis-preferred responses are generated from either corrupted images or misleading prompts. To further self-improve reasoning on the extracted visual information, we let the model reuse a small portion of existing instruction-tuning data and append its self-generated image descriptions to the prompts. We validate the effectiveness of STIC across seven different benchmarks, demonstrating substantial performance gains of $4.0\%$ on average while using $70\%$ less supervised fine-tuning data than the current method. Further studies investigate various components of STIC and highlight its potential to leverage vast quantities of unlabeled images for self-training. Code and data are made publicly available on GitHub.

## 1 Introduction

In recent years, we have witnessed remarkable advancements in large language models (LLMs), such as GPT-4 (OpenAI, 2023a) and the LLaMA family (Touvron et al., 2023a,b). The increasing importance of processing multimodal inputs, including images and text, has significantly driven progress in vision language models (Radford et al., 2021; Jia et al., 2021b; Goel et al., 2022). Leveraging the powerful language understanding and generation capabilities of LLMs, researchers have advanced vision language models into large vision language models (LVLMs). This enhancement is achieved by integrating LLMs with image encoders (Radford et al., 2021; Li et al., 2023a), which were pre-trained on large-scale image-text pairs to ensure alignment between the two domains. For instance, LLaVA (Liu et al., 2023b) integrates a vision encoder from CLIP (Radford et al., 2021) with the LLM Vicuna (Chiang et al., 2023b), which is further fine-tuned on carefully constructed vision-language instructional datasets to activate the model's perception capability of capturing the vision information according to different queries. This recent development has substantially expanded the requirement for large-scale instruction fine-tuning data for LVLMs (Gao et al., 2023b; Bai et al., 2023; Chen et al., 2023b; Gao et al., 2024; Anthropic, 2024; McKinzie et al., 2024).

---

[*]Equal contribution.

[†]Contribution statement is provided after Section 7.

38th Conference on Neural Information Processing Systems (NeurIPS 2024).

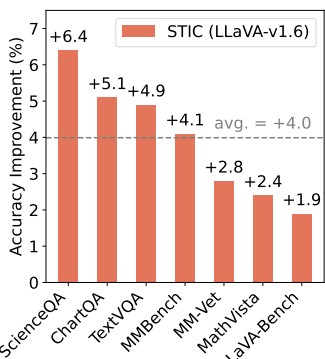
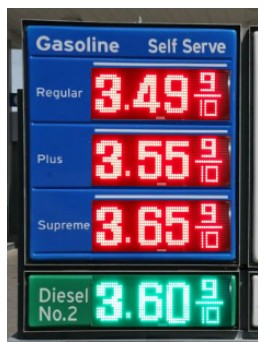

**Query:** How many gallons of supreme gasoline can I get with $50?

**Base (LLaVA-v1.6 7B):**
Based on the current gas prices displayed on the sign, you can get **approximately 3.65 gallons** of supreme gasoline with $50.

**STIC (LLaVA-v1.6 7B):**
With $50, you can get **approximately 13.69 gallons** of supreme gasoline, as indicated by the price of **$3.65 per gallon on the sign**.

Figure 1: **Left**: Accuracy improvement of our method, STIC, compared to the original LLaVA-v1.6 (Liu et al., 2024) on seven benchmarks. **Right**: Response examples from the original LLaVA-v1.6 and STIC (LLaVA-v1.6), which enhances image comprehension and subsequent reasoning capabilities.

While LVLMs have shown promising results, a key challenge lies in the acquisition of high-quality fine-tuning data. Obtaining human-curated content at scale is often prohibitively expensive, especially for multi-modal data. Many recent studies resort to GPT-4V (OpenAI, 2023b) for generating or labeling high-quality vision-language fine-tuning data. However, this approach does not significantly reduce the cost (Liu et al., 2023b; Wu et al., 2024). For instance, using GPT-4V to generate $6k$ image descriptions with $1k$ tokens per output would cost approximately $200. There remains a pressing need for cost-effective methods to gather fine-tuning data to further enhance LVLMs.

To tackle the data acquisition bottleneck in multi-modality, we propose **S**elf-**T**raining on **I**mage **C**omprehension (**STIC**). Our method is inspired by the recent success of self-training (Chen et al., 2024; Yuan et al., 2024; Fränken et al., 2024; Rosset et al., 2024) in LLMs, which leverages self-generated data to improve their downstream performance. However, different from the text-only domain, the unique vision modality of LVLMs introduces new challenges, as LVLMs must understand the input image content before reasoning and responding to any related textual queries about the image. Therefore, the proposed STIC approach is a novel two-stage self-training method that targets both *image perception* and *reasoning over images and texts*.

The overall framework is summarized in Figure 2. STIC specifically emphasizes the **image comprehension self-training** of LVLMs where the model generates its own preference dataset focused on image description. The self-generated *dispreferred response* is obtained by gathering model responses from either (1) prompts likely to elicit inaccurate responses or (2) corrupted images. The *preferred responses* are collected via a detailed prompt that guides the model through a step-by-step image description process. Figure 3 shows examples of such generated responses. During fine-tuning, we consider a DPO loss (Rafailov et al., 2023) with an additional regularized term explicitly emphasizing the preferred response. Lastly, we allow the model to self-improve its reasoning ability based on its own extracted image information by reusing a small amount of existing instruction fine-tuning data and appending its self-generated image descriptions to the prompts. We refer to this second stage as **description-infused fine-tuning**. Notably, our STIC approach *does not require pre-labeled information of the images*, which contrasts to the recent works that rely on such information for constructing vision-language preference data (Zhou et al., 2024).

To demonstrate the effectiveness of STIC, we conduct extensive experiments on seven vision-language benchmarks, including ScienceQA (Lu et al., 2022), TextVQA (Singh et al., 2019), ChartQA (Masry et al., 2022), LLaVA-Bench (Liu et al., 2023a), MMBench (Liu et al., 2023c), MM-Vet (Yu et al., 2023), and MathVista (Lu et al., 2024). These benchmarks encompass scientific reasoning, math reasoning, optical character recognition (OCR), and conversation capabilities based on vision inputs, spanning various image sources such as natural, chart, and text-rich images. We employ LLaVA-v1.6 (Liu et al., 2024) as the primary base LVLM for our experiments and unitize $6k$ images from MSCOCO (Lin et al., 2014) to construct the image description preference data. As depicted in Figure 1, STIC achieves consistent and significant performance improvements across these benchmarks, with an average accuracy gain of **4.0%** over the base LVLM and a notable gain of **6.4%** on ScienceQA. We also provide an example of the different responses from the original LVLM and STIC in Figure 1, where STIC successfully identifies the key visual information and accurately reason with it. These results demonstrate the remarkable effectiveness of our image comprehension self-training approach in enhancing the visual perception capabilities of LVLMs.

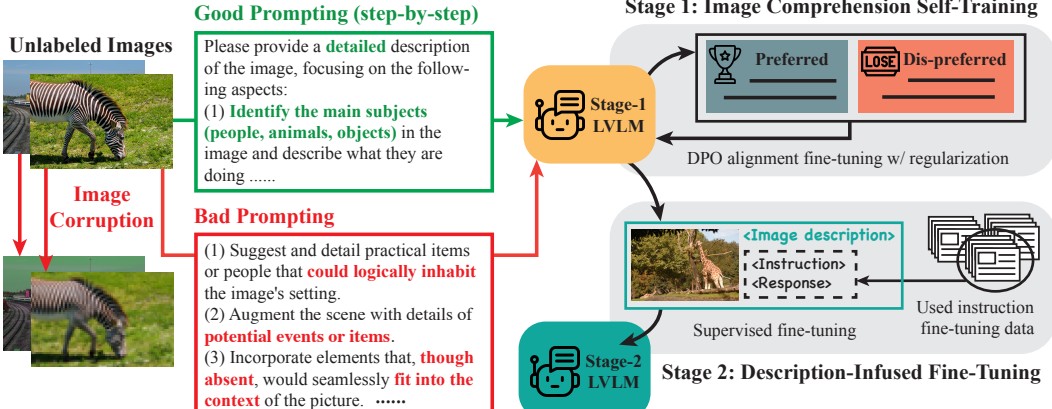

Figure 2: Framework overview of STIC, a two-stage self-training algorithm focusing on the image comprehension capability of the LVLMs. In Stage 1, the base LVLM self-constructs its preference dataset for image description using well-designed prompts, poorly-designed prompts, and distorted images. In Stage 2, a small portion of the previously used SFT data is recycled and infused with model-generated image descriptions to further fine-tune the base LVLM.

In addition, we explore the benefits of the various components of STIC. First, based on the description-infused fine-tuning stage that enhances the model's reasoning ability with self-generated description, we show that further letting the model describe the image before responding to a query provides further improved reasoning capability. This results in a notable improvement of $2.8\%$ on ScienceQA and $1.1\%$ on average as compared to direct responses to queries (Table 2). Moreover, we examine the impact of self-generated dispreferred responses, from either bad prompting or image corruption. By excluding these dispreferred responses and conducting SFT solely with preferred responses, we observed a performance decrease of $2.5\%$ on average across three benchmarks as compared to STIC with the preference data (Table 3). This highlights the importance of the negative samples in the self-constructed preference data by STIC. We also assess the scalability of our self-training scheme. By increasing the amount of generated preference data from $6k$ to $12k$, we show an even further improvement of STIC from $1.9\%$ to $3.1\%$ on LLaVA-Bench (Figure 6). This result suggests that STIC holds considerable potential for leveraging vast quantities of unlabeled images for self-training, given the immense availability of unlabeled image data. Lastly, our t-SNE visualization analysis shows that the closer the distribution between MSCOCO images, which we use for preference data construction, to images in downstream tasks, the more likely STIC results in higher performance gains (Figure 7).

The main contributions of this work are summarized as follows:

- We propose STIC, a novel two-stage self-training approach for LVLMs that focuses on enhancing their image comprehension capabilities by generating a preference dataset for image description without relying on pre-labeled image information.
- Through extensive experiments on seven diverse benchmarks, STIC demonstrates significant performance gains over the base LVLM, achieving an average accuracy gain of $4.0\%$.
- We explore the benefits of various components of STIC, highlighting its potential to leverage vast quantities of unlabeled images for self-training.

## 2   Related Work

**Vision language models (VLMs).** VLMs (Tan and Bansal, 2019; Li et al., 2019, 2020; Kim et al., 2021; Wang et al., 2022b; Bao et al., 2022; Wang et al., 2022a; Alayrac et al., 2022; Li et al., 2023b; Chen et al., 2022; Jia et al., 2021a; Shen et al., 2022; Singh et al., 2021), processing both images and text, are pivotal in a wide range of multimodal understanding and reasoning tasks, capable of generating text or encoding multimodal representations. These models have shown increasing proficiency in visual perception and textual reasoning, and are also capable of following complex instructions (OpenAI, 2023b; Team et al., 2023). Recent advancements in the field have been propelled by the availability of open-source large language models (LLMs) (Touvron et al., 2023a,b; Jiang et al., 2023) and innovative image encoders (Radford et al., 2021; Li et al., 2022). For instance, LLaVA (Liu et al., 2023b) combines a vision encoder from CLIP (Radford et al., 2021) with the Vicuna LLM (Chiang et al., 2023b), and has been further fine-tuned on vision-language instruction-

| Task: Explain what is depicted in the photograph. |
| --- |

*Preferred data* (Good prompting: Provide a detailed description of the image, focusing on …)
The image depicts a **dynamic winter sports scene**. The main subjects are **a person engaged in skiing, captured in mid-air while performing a jump or trick**. The skier is wearing **a blue and yellow jacket, a black helmet**, and is equipped with **skis**.....

**(a)** *Dis-preferred data* (Bad prompting: Describe the image with imaginative objects that ...)
...... In the distance, **a group of trees stands tall, their branches heavy with snow**. ...... Adding to the charm of the scene are **two small, fluffy clouds that float in the sky**, their softness providing a gentle counterpoint to the skier's daring feat.

**(b)** *Dis-preferred data* (Normal prompting + Image corruption: Color jitter.)
...... The skier, clad in a **vibrant purple jacket** and blue pants, ...... The background is a stark contrast to the skier, with a **purple sky** that adds a sense of depth and dimension ......

**(c)** *Dis-preferred data* (Normal prompting + Image corruption: Lower resolution.)
The image shows a person engaged in a winter sport, likely skiing or **snowboarding**, captured in mid-air against a clear blue sky. The individual is wearing a blue and yellow suit, ...... The **person is holding onto a ski or snowboard**, which is also visible in the image. The **motion blur effect** ......

Figure 3: Examples of the self-constructed preference data in STIC.

following datasets. The recent development of LVLMs has significantly expanded the scale and diversity of VL instruction-following data, including models such as LLaMA-Adapter-V2 (Gao et al., 2023b), Qwen-VL (Bai et al., 2023), InternVL (Chen et al., 2023b), InstructBLIP (Dai et al., 2024), SPHINX-X (Gao et al., 2024), Claude-3 (Anthropic, 2024), MM1 (McKinzie et al., 2024), and Grok-1.5V (xAI, 2024). In this work, we focus on enhancing the visual perception and mathmatical reasoning capabilities of LVLMs by efficiently aligning them with purely unsupervised data.

**Alignment fine-tuning.** Subsequent to supervised fine-tuning (SFT), alignment fine-tuning has emerged as a prominent approach to further enhance the performance of LLMs by aligning them with human preferences (Ouyang et al., 2022; Casper et al., 2023). Early efforts utilized on-policy reinforcement learning (RL) methods, such as proximal policy optimization (PPO) (Schulman et al., 2017), to train a reward model based on preference data (Bai et al., 2022; Touvron et al., 2023a). With the notable introduction of direct policy optimization (DPO) (Rafailov et al., 2023), a new line of research emphasizes direct learning from human preferences without relying on an explicit reward model (Zhao et al., 2023; Azar et al., 2024; Ethayarajh et al., 2024; Zheng et al., 2024). Another prominent direction is iterative preference fine-tuning, which has proven effective in enhancing model performance by repeatedly optimizing on newly generated preference pairs in each iteration (Adolphs et al., 2023; Xu et al., 2023; Xiong et al., 2023; Pang et al., 2024). While substantial research has focused on alignment fine-tuning for LLMs, efforts to adapt these techniques for LVLMs have been significantly limited. Initial attempts involve constructing preference datasets using human-labeled data (Sun et al., 2023) or GPT-4 generations for fine-tuning with a DPO loss (Zhou et al., 2024). Concurrent works (Pi et al., 2024; Zhou et al., 2024) begin to focus on generating preference dataset of LVLMs, while our method distinguishes itself with the unique preference prompt set.

**Self-training.** Traditional self-supervised training schemes (He et al., 2019; Xie et al., 2020; Wei et al., 2020; Zoph et al., 2020; Sohn et al., 2020; Ghiasi et al., 2021; Kang et al., 2023) leverage trained models to generate labels for unlabeled data and incorporate these self-labeled examples into training as a form of data augmentation. These frameworks primarily focus on self-supervised representation learning of vision models. While both classical self-training schemes and our approach share the fundamental goal of effectively utilizing unlabeled data to enhance model performance, our method differs in its focus on vision LLMs, maintaining an LLM as the backbone architecture. Rather than optimizing image representations, our approach aims to generate synthetic data that enables the LLM to produce higher-quality responses to image queries.

## 3 Problem Setting and Preliminaries

**Notation.** We use lower case letters and lower case bold face letters to denote scalars and vectors. We use the symbol $p$ to represent the probability of an LLM's response. And we denote the sequence of tokens generated from the LLM before the $t$-th token as $\mathbf{y}_{<t} = [y_1, \ldots, y_{t-1}]$ for $t > 1$.

**Generative vision language models.** LVLM typically consists of three components: a vision encoder $f(\cdot)$, a projection network $g(\cdot)$, and an LLM $p_{\boldsymbol{\theta}}$ parameterized by $\boldsymbol{\theta}$. The model processes an image input $\mathbf{e}$ along with a text sequence $\mathbf{x} = [x_1, \ldots, x_n]$ as the prompt to generate a corresponding

response $\mathbf{y} = [y_1, \ldots, y_m]$, where $x_i$ and $y_j$ represent individual tokens from the vocabulary of the LLM. The image is therefore converted into visual tokens within the language token space by the vision encoder and the projection network, producing $\mathbf{v} = [v_1, \ldots, v_k] = f \circ g(\mathbf{e})$. The response $\mathbf{y}$ is then considered as a sample from the conditional probability distribution $p_{\boldsymbol{\theta}}(\cdot|\mathbf{v}, \mathbf{x})$. As a Markov process, the conditional probability distribution $p_{\boldsymbol{\theta}}(\mathbf{y}|\mathbf{v}, \mathbf{x})$ can be decomposed as

$$p_{\boldsymbol{\theta}}(\mathbf{y}|\mathbf{v}, \mathbf{x}) = \prod_{j=1}^{m} p_{\boldsymbol{\theta}}(y_j|\mathbf{v}, \mathbf{x}, \mathbf{y}_{<j}). \tag{3.1}$$

**Alignment fine-tuning.** To improve LLM alignment with human preferences, RL fine-tuning (Bai et al., 2022; Gao et al., 2023a) is typically employed after supervised fine-tuning (SFT). This process involves a reward function $r(\mathbf{x}, \mathbf{y})$ for a given sequence pair $(\mathbf{x}, \mathbf{y})$. The more preferred response $\mathbf{y}$ is expected to result in a higher reward $r(\mathbf{x}, \mathbf{y})$, where the corresponding objective is to maximize the following:

$$L(\boldsymbol{\theta}) = \mathbb{E}_{\mathbf{x}\sim\mathcal{D}, \mathbf{y}\sim p_{\boldsymbol{\theta}}(\cdot|\mathbf{x})}[r(\mathbf{x}, \mathbf{y})] - \lambda\mathbb{E}_{\mathbf{x}\sim\mathcal{D}}\text{KL}\big(p_{\boldsymbol{\theta}}(\cdot|\mathbf{x})||p_{\text{ref}}(\cdot|\mathbf{x})\big), \tag{3.2}$$

where $\mathbf{x} \sim \mathcal{D}$ is sampled from a given distribution $\mathcal{D}$ and the KL regularization term prevents the new model $p_{\boldsymbol{\theta}}$ from deviating too much from the reference model $p_{\text{ref}}$, with $\lambda > 0$ as the regularization parameter. Training the reward function is challenging in practice, but direct preference optimization (DPO) (Rafailov et al., 2023) simplifies this process using a predefined preference dataset $S_{\text{pref}} = \big\{(\mathbf{x}^{(i)}, \mathbf{y}_w^{(i)}, \mathbf{y}_l^{(i)})\big\}_{i\in[N]}$, where $\mathbf{y}_w^{(i)}$ denotes the preferred response and $\mathbf{y}_l^{(i)}$ denotes the dispreferred response given the same prompt $\mathbf{x}^{(i)}$. The objective function is then formulated as

$$L_{\text{DPO}}(\boldsymbol{\theta}, \boldsymbol{\theta}_{\text{ref}}) = \mathbb{E}_{(\mathbf{x}, \mathbf{y}_w, \mathbf{y}_l)\sim S_{\text{pref}}}\left[\ell\bigg(\lambda\log\frac{p_{\boldsymbol{\theta}}(\mathbf{y}_w|\mathbf{x})}{p_{\boldsymbol{\theta}_{\text{ref}}}(\mathbf{y}_w|\mathbf{x})} - \lambda\log\frac{p_{\boldsymbol{\theta}}(\mathbf{y}_l|\mathbf{x})}{p_{\boldsymbol{\theta}_{\text{ref}}}(\mathbf{y}_l|\mathbf{x})}\bigg)\right], \tag{3.3}$$

where $\ell(t) = \log(1 + \exp(-t))$ is the logistic loss function and $\boldsymbol{\theta}_{\text{ref}}$ is the reference model.

# 4 Our Method: STIC

In this section, we introduce STIC, a two-stage self-training algorithm designed to enhance image comprehension capabilities. The first stage constructs its own preference dataset and the second stage infuses the used SFT data with *self-generated* image descriptions for fine-tuning. Figure 2 presents the general framework of STIC. Notably, unlike recent work on fine-tuning algorithms (Sun et al., 2023; Zhou et al., 2024), STIC enables a base LVLM, such as LLaVA-v1.6 (Liu et al., 2024), to evolve from *self-generated* image captions, thus eliminating the need for additional supervised and preference data from human annotators or advanced teacher models. This approach *fundamentally* enhances image comprehension abilities and can be seamlessly applied to a wide range of vision-language reasoning tasks. We summarize STIC in Algorithms 1 and 2, and detail the process below.

**Stage 1: Image comprehension self-training.** The process begins with a self-constructed preference dataset from the base LVLM, which we aim to improve through fine-tuning. The dataset contains paired preference data for image descriptions:

- *Preferred* response: Model-generated image descriptions derived from well-crafted prompts with explicit reasoning steps.
- *Dispreferred* response: Model-generated descriptions resulting from either (1) corrupted image with low resolution or distorted color, or (2) "bad" prompts that cause the base model to hallucinate and describe elements that may not logically exist in the image.

The self-constructed preference dataset is used for the first-stage self-training using DPO (Rafailov et al., 2023) with an additional regularization term to further emphasize the preferred response, controlled by the hyperparameter $\alpha$. The regularized loss function is as follows:

$$L(\boldsymbol{\theta}, \boldsymbol{\theta}_{\text{ref}}) = \mathbb{E}_{(\mathbf{x}, \mathbf{y}_w, \mathbf{y}_l)\sim S}\left[\ell\bigg(\lambda\log\frac{p_{\boldsymbol{\theta}}(\mathbf{y}_w|\mathbf{x})}{p_{\boldsymbol{\theta}_{\text{ref}}}(\mathbf{y}_w|\mathbf{x})} - \lambda\log\frac{p_{\boldsymbol{\theta}}(\mathbf{y}_l|\mathbf{x})}{p_{\boldsymbol{\theta}_{\text{ref}}}(\mathbf{y}_l|\mathbf{x})}\bigg) - \alpha\log p_{\boldsymbol{\theta}}\big(\mathbf{y}_w|\mathbf{x}\big)\right]. \tag{4.1}$$

The use of an explicit loss term for positive examples can be similarly found in previous studies on contrastive learning (Chen et al., 2021; Chen and He, 2021; Chen et al., 2023a) and more recently in preference fine-tuning (Pang et al., 2024). Specifically, Chen et al. (2023a) demonstrated in the context of contrastive learning that a regularization term applied to positive samples provably enhances the model's ability to differentiate between positive and negative samples. As demonstrated in our experiments in Section 6, the LVLM after Stage 1 has shown notable improvement in downstream

---

**Algorithm 1** STIC (Stage 1: image comprehension self-training)

---

**Input:** Unlabeled image dataset: $\{\mathbf{v}^{(i)}\}_{i \in [N]}$. Image captioning prompt set: $P = \{\mathbf{x}^{(i)}\}_{i \in [M_1]}$.
Hallucination prompt set: $P_{\text{hallu}} = \{\mathbf{x}_{\text{hallu}}^{(i)}\}_{i \in [M_2]}$. Image corruption $h(\cdot)$. Well-curated caption-
ing prompt: $\mathbf{x}_g$. LVLM parameterized by $\boldsymbol{\theta}_0$: $p_{\boldsymbol{\theta}_0}$.
Let self-training dataset $D = \{\}$.
**for** $i = 1, \ldots N$ **do**
    Randomly sample a number $n \in (0, 1)$.
    Randomly sample $\mathbf{x} \sim \{\mathbf{x}^{(i)}\}_{i \in [M]}$.
    Generate preferred response $\mathbf{y}_g \sim p_{\boldsymbol{\theta}_0}(\cdot | \mathbf{v}^{(i)}, \mathbf{x}_g)$.
    **if** $n < 0.5$ **then**
        Randomly sample bad prompt $\mathbf{x}_b \sim P_{\text{hallu}}$.
        Generate dispreferred response $\mathbf{y}_b \sim p_{\boldsymbol{\theta}_0}(\cdot | \mathbf{v}^{(i)}, \mathbf{x}_b)$.
    **else**
        Corrupt the image input $\mathbf{v}_b^{(i)} = h(\mathbf{v}^{(i)})$.
        Generate dispreferred response $\mathbf{y}_b \sim p_{\boldsymbol{\theta}_0}(\cdot | \mathbf{v}_b^{(i)}, \mathbf{x})$.
    **end if**
    Add $(\mathbf{x}, \mathbf{y}_g, \mathbf{y}_b)$ to $D$.
**end for**
Update $\boldsymbol{\theta}_1 = \operatorname{argmin}_{\boldsymbol{\theta} \in \Theta} \sum_{(\mathbf{x}, \mathbf{y}_g, \mathbf{y}_b) \in D} \left[ \ell \left( \lambda \log \frac{p_{\boldsymbol{\theta}}(\mathbf{y}_g | \mathbf{x})}{p_{\boldsymbol{\theta}_0}(\mathbf{y}_g | \mathbf{x})} - \lambda \log \frac{p_{\boldsymbol{\theta}}(\mathbf{y}_b | \mathbf{x})}{p_{\boldsymbol{\theta}_0}(\mathbf{y}_b | \mathbf{x})} \right) - \alpha \log p_{\boldsymbol{\theta}}(\mathbf{y}_g | \mathbf{x}) \right]$.

**Output:** $\boldsymbol{\theta}_1$.

---

vision-language reasoning tasks, confirming that the enhanced visual comprehension ability directly benefits the model performance and its multimodal reasoning ability.

**Prompt design.** Our prompt design for the well-crafted prompt focuses on quality and diversity. We use GPT-4 to generate and sample multiple initial prompts, which are then refined through human filtering. To ensure effectiveness, we test these prompts on MSCOCO samples, verifying their ability to produce well-structured and relevant responses from the model. The bad prompts are sampled from GPT-4 and, in contrast, designed to elicit inaccurate descriptions by setting up a slightly different task (describe objects that would logically exist in the image) for the model. We thus work under the assumption that responses generated from prompts that have differences in human preference lead to responses of the same preference with high probability. The key is that the discrepancy between good and bad prompts should result in pairs of responses that share the same implicit preference with high probability, which is sufficient for effective DPO training.

**Stage 2: Description-infused fine-tuning.** In the second stage, we further fine-tune the self-trained LVLM to leverage self-generated high-quality image descriptions for instruction-following tasks, and thus help ground its reasoning ability on self-generated descriptions. To achieve this, we randomly select a small subset of data from the model's instruction fine-tuning dataset already used during SFT. We then infuse the instructions in this subset with model-generated image descriptions as follows:

```
Image description:  {model description}
<original instruction>
```

The original ground-truth completions remain unchanged. We then fine-tune the LVLM for one epoch on this small description-infused subset. This fine-tuning step ensures that the model effectively integrates visual information into its responses, thereby enhancing its ability to handle a variety of vision-language reasoning tasks.

**Describe and Respond.** During inference, optionally, we can let the model self-augment its prompt for downstream vision-language reasoning tasks by describing the image before answering. Rather than generating an immediate response, we first elicit an image description, which is then concatenated with the original question to produce a more informed answer.

## 5 Experiments

In this section, we present the experiment results of STIC across seven visual question answering (VQA) benchmarks. We demonstrate that STIC effectively and substantially improves LVLM's performance across different VQA tasks using a self-constructed preference dataset without labels.

---
**Algorithm 2** STIC (Stage 2: description-infused fine-tuning)
---
**Input:** Instruction-following dataset already used for fine-tuning the target LVLM model: $\{\mathbf{v}^{(i)}, \mathbf{x}^{(i)}, \mathbf{y}^{(i)}\}_{i\in[m]}$. Image description prompt set: $P = \{\mathbf{x}_{\text{des}}^{(i)}\}_{i\in[M_1]}$. LVLM parameterized by $\boldsymbol{\theta}_1$ after self-training: $p_{\boldsymbol{\theta}_1}$.

Let description-infused dataset $D_{\text{des}} = \{\}$.

**for** $i = 1, \ldots m$ **do**

    Randomly sample $\mathbf{x}_{\text{des}} \sim \{\mathbf{x}_{\text{des}}^{(i)}\}_{i\in[M]}$.

    Generate model image description $\mathbf{y}_{\text{des}} \sim p_{\boldsymbol{\theta}_t}(\cdot|\mathbf{v}^{(i)}, \mathbf{x}_{\text{des}})$.

    Add $\left([\mathbf{y}_{\text{des}}, \mathbf{x}^{(i)}], \mathbf{y}^{(i)}\right)$ to $D_{\text{des}}$.

**end for**

Update $\widehat{\boldsymbol{\theta}} = \text{argmin}_{\boldsymbol{\theta}\in\boldsymbol{\Theta}} \sum_{(\mathbf{x},\mathbf{y})\in D_{\text{des}}} \ell\left(\log p_{\boldsymbol{\theta}}(\mathbf{y}|\mathbf{x})\right)$.

**Output:** $\widehat{\boldsymbol{\theta}}$.

---

## 5.1 Experiment Setup

**Model and datasets.** In experiments, we consider `llava-v1.6-mistral-7b` (Liu et al., 2023a) as our base model for self-training with model generated preference data. We additionally consider `llava-v1.5-7b` (Liu et al., 2023a) based on Vicuna-7B (Chiang et al., 2023b) to directly compare with one concurrent baseline POVID (Zhou et al., 2024). A detailed discussion with POVID can be found in Appendix C.3. We follow the optimization process described in Section 4 for self-training on image description in Algorithm 1 and description-infused fine-tuning in Algorithm 2 to achieve improved downstream performances. For the self-constructed preference dataset, we gather **6$k$ unlabeled image data** randomly sampled from the MSCOCO dataset (Lin et al., 2014) and specifically the `train2014` split for its high-quality images popularly used for pre-training and fine-tuning. In the second stage, we randomly subsample **5$k$ used instruction fine-tuning data** from LLaVA's SFT data to construct the description-infused fine-tuning data with model-generated image descriptions. Lastly, we use low-rank adaptation (LoRA) fine-tuning (Hu et al., 2021) instead of full fine-tuning for efficient computation. We defer the detailed prompts and corruptions to Appendix B.

**Evaluation.** We consider the widely used benchmarks for LVLM evaluation across different domains including: ScienceQA (Lu et al., 2022), TextVQA (Singh et al., 2019), ChartQA (Masry et al., 2022), LLaVA-Bench (Liu et al., 2023a), MMBench (Liu et al., 2023c), MM-Vet (Yu et al., 2023), and MathVista (Lu et al., 2024). Specifically, ScienceQA focuses on scientific question answering and MathVista focuses on math reasoning with visual information. TextVQA consists of images with text-rich contents and ChartQA with visual charts. Lastly, LLaVA-Bench, MMBench, and MM-Vet are three recent benchmarks to comprehensively evaluate a model's capabilities in a wide range of tasks and evaluation criteria. We use the evaluation scripts provided by LLaVA (Liu et al., 2023a) to obtain the results for both our base model and after using STIC to ensure a fair comparison.

## 5.2 Main Results

Table 1: Performance of STIC compared with the original LVLM model across vision-language reasoning tasks. For LLaVA-v1.5 (Vicuna 7B), we directly report the values in the paper of POVID, and "–" indicates an unreported value.

| Model | ScienceQA | TextVQA | ChartQA | LLaVA-Bench | MMBench | MM-Vet | MathVista |
|---|---|---|---|---|---|---|---|
| InstructBLIP (7B) | 60.5 | 50.1 | – | 60.9 | 36.0 | 26.2 | 25.3 |
| mPLUG-OWL2 (7B) | 64.5 | 54.3 | – | 59.9 | 64.5 | 36.2 | 22.2 |
| LLaVA-v1.5 (7B) | 66.8 | 58.2 | 6.3 | 65.4 | 64.3 | 31.1 | 25.1 |
| w/ POVID | 68.8 | – | – | 68.7 | 64.9 | 31.8 | – |
| w/ STIC | **69.5** | **61.4** | **6.6** | **68.9** | **65.3** | **32.6** | **27.2** |
| LLaVA-v1.6 (7B) | 68.9 | 60.3 | 36.4 | 77.3 | 63.7 | 42.2 | 34.6 |
| w/ STIC | **75.3** | **65.2** | **41.5** | **79.2** | **67.8** | **45.0** | **37.0** |

We present our main results in Table 1 and detail the benchmark performances of STIC (LLaVA-v1.6 7B) on MMBench and MM-Vet in Figure 4. In Appendix B, we present detailed results for MMBench in Table 7 and MM-Vet in Table 8. Our results show a consistent and significant improvement of STIC over the original models (LLaVA-v1.5 and LLaVA-v1.6) across all seven datasets. This improvement is achieved using only self-constructed preference data and a small portion of the model's SFT dataset, which had already been used for fine-tuning the original model.

On average, STIC improves LLaVA-v1.5 by 1.7%, increasing from 45.3% to 47.0%, and LLaVA-v1.6 by a notable score of 4.0%, increasing from 54.7% to 58.7%. The improvement is comprehensive, as detailed in Tables 7 and 8, where STIC consistently enhances performance across all evaluation tasks and targets. Moreover, while STIC improves both LLaVA-v1.5 and LLaVA-v1.6, a more significant improvement is observed in the more advanced model, LLaVA-v1.6. This trend suggests that the extent of self-improvement could be correlated with the model's inherent capabilities.

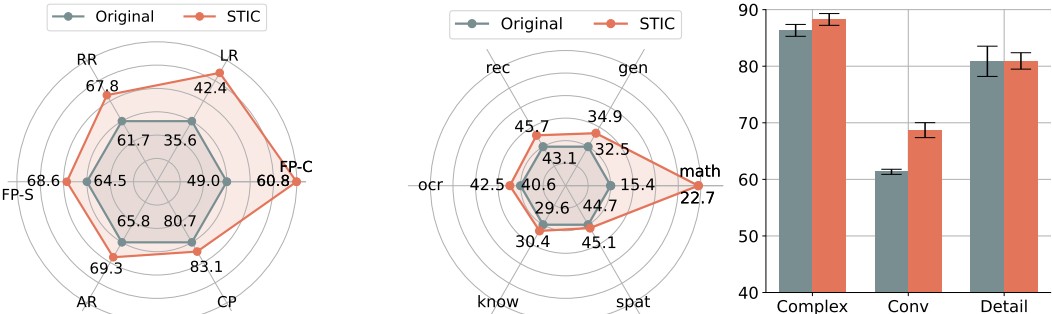

Figure 4: Accuracy improvement of STIC compared to the base LLaVA-v1.6 model across different tasks in **Left**: MMBench, where the original performances are re-scaled to 60 in plotting and STIC accordingly with the same coefficient for each task. **Middle**: MM-Vet, where the performances of the original model are re-scaled to 40 and STIC accordingly. **Right**: LLaVA-Bench, where we report the error bars over three independent runs due to the randomness of GPT-4 evaluation.

## 6 Ablation Studies and Discussions

In this section, we conduct ablation studies on the key components of STIC to demonstrate their importance and effectiveness. Additionally, we examine the image distribution of our self-training data (MSCOCO) alongside the image distributions of benchmark datasets, revealing a positive correlation between performance gains and similarity in image distributions.

**Effectiveness of describe-and-respond (DaR) prompting.** We assess the significance of the fine-tuning process in STIC by comparing it to the approach of directly allowing the base LVLM to describe an image and then respond to a query with a self-augmented prompt, which we refer to as the describe-and-respond (DaR) prompting method. As indicated in Table 2, applying DaR to the base LVLM yields mixed results, with performance improvements on some datasets and degradation on others, resulting in an overall average drop of 2.3%. In contrast, when DaR is combined with the fine-tuning process of STIC, it leads to a further average enhancement of 1.1% and a notable 2.8% on ScienceQA. This demonstrates the synergistic effect of DaR and the fine-tuning process in STIC. Additionally, it is worth noting that STIC achieves a substantial average improvement of 2.8% even without the DaR prompting method, compared to the base LVLM.

Table 2: Test performance of STIC based on `llava-v1.6-mistral-7b`. We investigate the benefit of DaR as a prompting method toward the base LVLM model as compared to the benefit on STIC.

| Method | DaR | ScienceQA | TextVQA | ChartQA | LLaVA-Bench | MMBench | MM-Vet | MathVista | Average |
|--------|-----|-----------|---------|---------|-------------|---------|--------|-----------|---------|
| Original | ✗ | 68.9 | 60.3 | 36.4 | 77.3 | 63.7 | 42.2 | 34.6 | 54.8 |
|  | ✓ | 69.9 | 56.6 | 34.6 | 78.5 | 50.7 | 42.3 | 34.7 | 52.5 |
| w/ STIC | ✗ | 72.5 | 63.4 | 39.3 | 78.4 | **68.7** | **45.7** | 35.2 | 57.6 |
|  | ✓ | **75.3** | **65.2** | **41.5** | **79.2** | 67.8 | 45.2 | **37.0** | **58.7** |

**Progression of stages.** In Figure 5, we illustrate the sequential improvement in performance of STIC on ScienceQA. While stage 1 focuses exclusively on enhancing the perception capabilities of the LVLM, it still notably improves performance on downstream VQA tasks. Building on the improved image comprehension achieved in stage 1, stage 2 introduces an enhanced reasoning process that utilizes the model's self-generated image descriptions and results in an even more significant gain. This enhancement further enables the model to self-augment its prompts with Describe and Respond (DaR), resulting in total the substantial performance gains of 6.4% observed.

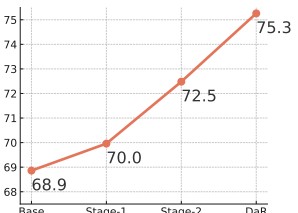

Figure 5: Progression of stages in STIC.

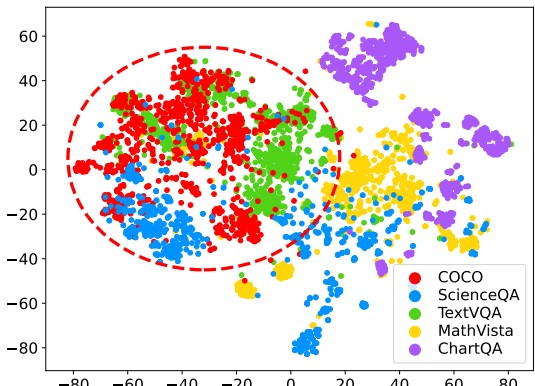

Figure 7: t-SNE visualization of images from MSCOCO and four benchmarks, each sampling $1k$.

**The role of dispreferred samples in STIC.** To understand the importance of dispreferred samples in STIC, we conduct an ablation study using `llava-v1.6-mistral-7b` as the base LVLM. We remove the negative examples from the preference data and only use the positive samples for supervised fine-tuning (SFT), effectively creating an SFT version of STIC. Table 3 shows that omitting the dispreferred samples even leads to a performance drop of $0.6\%$ on LLaVA-Bench, while failing to provide equally significant improvement as STIC with preference data. This highlights the crucial role of negative examples in aligning preferences and enabling the model to distinguish between high-quality and low-quality responses. By leveraging both positive and negative examples, STIC effectively improves the model's performance and generates more preferred outputs.

Table 3: Test performance of STIC if we remove negative examples and use positive ones to perform SFT in Stage 1.

| Model | ScienceQA | TextVQA | LLaVA-Bench |
|---|---|---|---|
| Original | 68.9 | 60.3 | 77.3 |
| w/ STIC (positive) | 71.8 | 63.7 | 76.7 |
| w/ STIC | **75.3** | **65.2** | **79.2** |

**Scaling law of STIC.** We explore the scaling law of STIC by expanding the preference data in Stage 1. Using the LLaVA-Bench benchmark as a case study, we scale up the preference data from $6k$ to $12k$ MSCOCO images. As depicted in Figure 6, there is an obvious gain on the LLaVA-Bench from $1.9\%$ to $3.1\%$. This finding demonstrates that STIC can effectively leverage larger amounts of unlabeled image data and presents a cost-effective solution to the challenge of acquiring high-quality vision-language data.

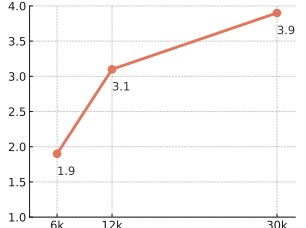

Figure 6: Scaling law in STIC.

**Correlation between image distribution and performance gains.** To gain further insight into the effectiveness of STIC across different benchmarks, we conducted a t-SNE visualization analysis comparing the image distributions of MSCOCO, which we used for preference data construction, with those of four benchmarks: ScienceQA, TextVQA, MathVista, and ChartQA (Figure 7). Our analysis revealed a general trend: the greater the overlap between the MSCOCO image distribution and that of a benchmark, the higher the performance gain achieved by STIC on that benchmark. This observation held true for ScienceQA and TextVQA, which exhibited substantial distributional overlap with MSCOCO and yielded the highest performance gains of $6.4\%$ and $4.9\%$, respectively. Conversely, MathVista, with its diverse image types and limited overlap with MSCOCO, saw a more modest gain of $2.4\%$. Interestingly, ChartQA was an outlier, achieving a high gain of $5.1\%$ despite minimal overlap with MSCOCO, suggesting that the improved image comprehension from STIC played a fundamental role in understanding and reasoning about the charts. Detailed per-benchmark visualizations and discussions are provided in Appendix C.2.

**Diversity in image distribution.** Based on the observation on the effect of image distribution in the final performance, we further utilize the Vision Flan dataset (VFLAN[†]) for stage 1 image comprehension self-training. This dataset includes images from 191 diverse vision tasks, providing a broader spectrum of image types. We ensured a fair comparison by maintaining the same sample

---

[†] https://huggingface.co/datasets/Vision-Flan/vision-flan_191-task_1k

Table 4: Performance of STIC on different stage-1 training images compared with the original LVLM model LLaVA-v.16 (Mistral 7B) across vision-language reasoning benchmarks.

| Model | Data | LLaVA-Bench | | | | MM-Vet | | | | | | | MMBench |
|---|---|---|---|---|---|---|---|---|---|---|---|---|---|
| | | Complex | Conv | Detail | **All** | Rec | Ocr | Know | Gen | Spat | Math | **Total** | **All** |
| LLaVA-v1.6 (7B) | - | 87.4 | 61.3 | 77.8 | 77.3 | 43.1 | 40.6 | 29.6 | 32.5 | 44.7 | 15.4 | 42.2 | 63.7 |
| w/ STIC | COCO | 89.1 | 63.7 | 79.5 | 79.2 | 45.7 | 42.5 | 30.4 | 34.9 | 45.1 | 22.7 | 45.0 | 67.8 |
| w/ STIC | VFLAN | 92.8 | 68.4 | 77.9 | **81.9** | 45.7 | 43.0 | 31.0 | 36.2 | 45.1 | 26.5 | **45.1** | **68.3** |

Table 5: Performance of STIC compared with the original LVLM model LLaVA-v1.6 (Vicuna 13B) across vision-language reasoning tasks. Image data used for 13B model remain the same as what we used for the 7B model.

| Model | LLaVA-Bench | | | | MM-Vet | | | | | | | MMBench |
|---|---|---|---|---|---|---|---|---|---|---|---|---|
| | Complex | Conv | Detail | **All** | Rec | Ocr | Know | Gen | Spat | Math | **Total** | **All** |
| LLaVA-v1.6 (7B) | 87.4 | 61.3 | 77.8 | 77.3 | 43.1 | 40.6 | 29.6 | 32.5 | 44.7 | 15.4 | 42.2 | 63.7 |
| LLaVA-v1.6 (13B) | 94.0 | 73.8 | 78.7 | 84.5 | 52.2 | 47.1 | 38.8 | 45.2 | 42.7 | 26.9 | 48.9 | 70.6 |
| w/ STIC | 93.5 | **78.1**$_{(+4.3)}$ | 79.4 | **85.6**$_{(+1.1)}$ | **54.5** | **48.0** | **42.3**$_{(+3.5)}$ | **49.4**$_{(+4.2)}$ | 42.0 | 23.1 | **50.5**$_{(+1.6)}$ | **72.3**$_{(+1.7)}$ |

size (randomly sampled $6k$ images) and present the experimental results in Table 4. As shown, our approach improves consistently across different datasets, demonstrating its robustness and adaptability. Notably, the increased diversity of VFLAN led to further improvements in STIC, suggesting the potential for even greater enhancement with better sets of unlabeled images.

**Scalability.** To explore STIC's applicability to models with higher representation capacity, we conducted supplementary experiments using LLaVA-v1.6 (Vicuna-13B). Table 5 shows the detailed experiment results. We used the same images for STIC fine-tuning as in our experiments for LLaVA-v1.6 (Mistral-7B) to ensure fairness and the same set of hyperparameters. The improvements observed with LLaVA-v1.6 (Vicuna-13B) demonstrate that STIC is not only effective with smaller models but also scales well with larger or more capable LVLMs.

**Qualitative example.** In Figure 8, we show an example output of STIC. Despite the task being focused on mathematical reasoning, STIC enhanced the model's response by improving its image comprehension capabilities. While the original model merely identified one of the recognized numbers in the image as the final answer, the STIC fine-tuned model was able to interpret the meaning of each number, describe them accurately, and perform reasoning based on this understanding.

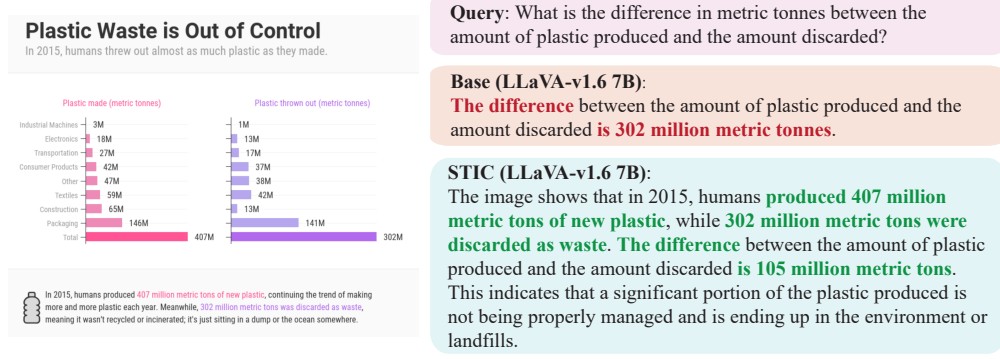

Figure 8: Response examples from original LLaVA-v1.6 and STIC (LLaVA-v1.6) in MM-Vet.

## 7 Conclusion

We introduce Self-Training on Image Comprehension (STIC), a novel self-training approach designed to enhance the image comprehension capabilities of large vision language models (LVLMs). Our method leverages a two-stage self-training process, creating a preference dataset for image descriptions from unlabeled images and refining reasoning abilities through description-infused fine-tuning. Extensive experiments across seven vision-language benchmarks demonstrated significant performance improvements, with an average accuracy gain of $4.0\%$, while reducing the need for supervised fine-tuning data by $70\%$. Our findings underscore the potential of STIC to harness vast quantities of unlabeled images, offering a cost-effective solution for advancing LVLMs.

The promising results demonstrated by STIC in enhancing the capabilities of 7B LVLMs suggest its potential applicability to larger models, such as those with 40B, and 100B parameters, if computational resources permit.

## Contribution Statement

The project began in January 2024 with Deng as part of Gu's group, collaborating with Lu from Chang's group. This project was in the line of research on self-training using synthetic data developed by Gu and Deng. Deng proposed the initial idea of this project, which was further developed with Lu. In February 2024, Yin joined the project. Both Gu and Chang offered early feedback to help shape the project. In March, Chang met with Deng, Lu, and Yin to discuss the initial experiment results and finalize the research plan. Chang continued supervising the project through one-on-one meetings with Lu. In April, Deng changed advisors and moved to Wang's lab, inviting Hu, Shen, Wang, and Zou to join the project. Hu and Shen contributed algorithmic improvements, while Wang and Zou provided detailed feedback on the experimental design and ablation studies. Deng and Lu primarily conducted the experiments and drafted the paper.

## Acknowledgments

We sincerely thank the anonymous reviewers for their helpful comments. Pan is supported by Bloomberg Data Science Ph.D. Fellowship, Qualcomm Innovation Fellowship and UCLA Dissertation Year Fellowship. Fan and Kai-Wei are supported by ONR grant N00014-23-1-2780, OptumLabs, and CISCO gift grants. Wei is supported by DARPA HR0011-24-9-0370, NSF 2200274, 2106859, 2312501, and NIH U54HG012517, U24DK097771. This paper is partially supported by Chan Zuckerberg Initiative. The views and conclusions contained in this paper are those of the authors and should not be interpreted as representing any funding agencies.

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

# A   Additional Related Work

**Self-improving language models.** High-quality data, including human-crafted and advanced AI generated content, has been demonstrated to significantly enhance the performance of LLMs on various tasks (Josifoski et al., 2023; Taori et al., 2023; Chiang et al., 2023a; Li et al., 2023c). Although, acquiring such high-quality data is often prohibitively expensive. To circumvent the costs associated with obtaining human-annotated or expertly curated data, researchers have shifted their focus to leveraging data generated by the target model itself, exploring ways of self-improvement (Chen et al., 2024; Yuan et al., 2024; Fränken et al., 2024; Rosset et al., 2024). Recent studies have also emphasized the rephrasing capabilities of LLMs, which either enhance their own response quality (Deng et al., 2023; Prasad et al., 2023) or augment synthetic data for self-supervised fine-tuning (Kim et al., 2023). To the best of our knowledge, our work is the first to explore the potential for self-improvement in LVLMs, specifically focusing on the vision modality and emphasizing the self-improvement of image comprehension capabilities.

# B   Experimental Details

**Perference data construction.** We consider randomly sampling from the following "bad" prompts as a means of generating dis-preferred examples.

- "Describe the image with imaginative objects that may exist in the scene."
- "Enrich the description by adding hypothetical objects or characters that could be part of the scene."
- "Suggest and detail practical items or people that could logically inhabit the image's setting."
- "Incorporate elements that, though absent, would seamlessly fit into the context of the picture."
- "Imagine and describe additional everyday objects or activities taking place just out of frame."
- "Augment the scene with details of potential events or items that are plausible."
- "Conceive of and detail natural elements, such as weather or animals, that could realistically enter the scene. Make the description affirmative."
- "Invent and incorporate details of practical tools, vehicles, or gadgets that could be expected in a similar scenario."

Given an input image, with $50\%$ chance, we generate the dis-preferred response using the "bad" prompt. Otherwise, we generate the dis-preferred response with a corrupted image either from color jittering or lower resolution. To generate the preferred response, we use the following step-by-step prompt:

- "Please provide a detailed description of the image, focusing on the following. Identify the main subjects (people, animals, objects) in the image and describe what they are doing. Describe the setting of the image. Is it indoors or outdoors? What kind of environment or location does it depict? What mood does the image convey? Are there any specific elements (such as lighting, weather, expressions) that contribute to this atmosphere? Describe the dominant colors and the overall composition. How do these elements affect the image's impact? Point out any details or symbols that might be relevant to understanding the image's meaning or context. If applicable, provide interpretations of what the image might represent or communicate."

The prompts used as instructions in DPO are listed below:

- "Illustrate the details of the picture."
- "Summarize the visual content presented."
- "Explain what is depicted in the photograph."
- "Outline the key elements captured in the image."
- "Detail the composition and subjects within the frame."
- "Convey the atmosphere and mood represented in the snapshot."
- "Interpret the scene shown in the image."
- "Identify and describe the main focal points in the visual."

In Stage 2, we prompt the LVLM with simple instructions like "Explain what is depicted in the photograph." to gather the image descriptions for Stage 2 fine-tuning.

**Task: Explain what is depicted in the photograph.**

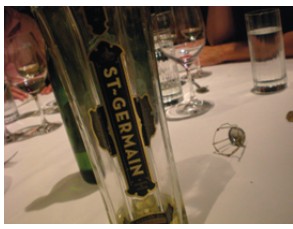

**Preferred (Good prompting)**
The image depicts a social setting, likely a dinner party or a gathering of some sort. The main subjects in the image are **a bottle of St-Germain liqueur and several wine glasses**. The bottle is prominently placed in the foreground, with its label facing the viewer, while the wine glasses are scattered around the table, some filled with wine.\n\n**The setting appears to be indoors, possibly a dining room or a similar space, as indicated by the presence of a dining table. The table is covered with a white tablecloth**, ......

**Dis-preferred (Bad prompting)**
In the image, there's a sense of a social gathering or a celebration. Just out of frame, you might imagine several everyday objects and activities:\n\n1. **A person is holding a glass of wine**, toasting to a special occasion.\n2. **A waiter is walking by, carrying a tray with appetizers or desserts**.\n3. A **guest** is engaged in a conversation with another person, sharing stories and laughter.\n4. A **person** is taking a photograph of the moment, capturing the joy and camaraderie.\n5. A host is standing near the **bar**, ensuring everyone has what they need. ......

Figure 9: Example of generated preference data, where the dis-preferred response is generated from bad prompting.

**Task: Explain what is depicted in the photograph**

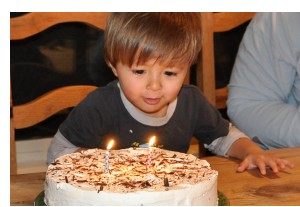

**Preferred (Good prompting)**
The image depicts a **young child**, likely a boy, seated at a **dining table** with a **birthday cake** in front of him. The child appears to be in the process of blowing out the **candles** on the cake, which are **lit and casting a warm glow on his face**. The child is wearing a **gray t-shirt** and has short, light-colored hair.\n\nThe setting is indoors, with a **wooden dining table and chairs** visible in the background. The environment suggests a domestic, homey atmosphere, possibly a dining room or a kitchen area.\n\nThe mood of the image is joyful ......

**Dis-preferred (Lower resolution)**
The image shows a close-up of a person, likely a child, looking at a cake with candles. The cake appears to be a birthday cake, and the child seems to be focused on it. The background is blurred, but it looks like an indoor setting with warm lighting. **The focus is on the child and the cake, making the background indistinct**.

Figure 10: Example of generated preference data, where the dis-preferred response is generated from images with lower resolution.

**Task: Interpret the scene shown in the image.**

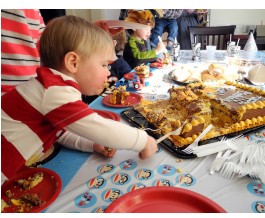

**Preferred (Good prompting)**
The image depicts a **joyful and festive scene**, likely a birthday party. The main subjects in the image are **a young child**, who appears to be the focus of the celebration, and several **other children and adults**. The child is seated at a table, engrossed in eating a piece of cake. ...... The setting is indoors, with the **table and chairs** arranged in a typical party layout. The table is covered with a **tablecloth featuring a cartoon character**, adding to the festive atmosphere. ...... The mood of the image is one of happiness and anticipation ......

**Dis-preferred (Color jitter)**
The image shows a child's birthday party. The child in the foreground is wearing a striped shirt and appears to be in the process of eating a piece of cake. **The cake is green** and has a \"Happy Birthday\" message on it, suggesting it's a birthday celebration. The child is using a plastic fork to eat the cake.\n\nIn the background, there are other children and adults, some of whom are also eating cake. ...... The overall atmosphere seems to be one of joy and celebration.

Figure 11: Example of generated preference data, where the dis-preferred response is generated from images with color jittering.

**Fine-tuning details.**    We train for 1 epoch in each stage, including the image comprehension self-training stage and the description-infused fine-tuning stage. We use the same hyperparameters for LoRA fine-tuning in both stages, with `lora_r = 128`, `lora_alpha = 256`, and `lora_target = all`. The fine-tuning hyperparameters for Stage 1 are presented in Table 6. The parameters remain the same for Stage 2 fine-tuning, with the only differences being a learning rate of $2e-5$ and a global batch size of $64$.

Table 6: Fine-tuning hyperparameters.

| | |
|---|---|
| Learning rate | $1e-7$ |
| Optimizer | `AdamW` |
| Global batch size | 4 |
| Regularization coefficient $\alpha$ | $1/1024$ |
| `mm_projector_lr` | $2e-5$ |
| `mm_projector_type` | `mlp2x_gelu` |
| `gradient_accumulation_steps` | 1 |
| `image_aspect_ratio` | `pad` |
| `group_by_modality_length` | `True` |
| `weight_decay` | 0 |
| `warmup_ratio` | 0.03 |
| `lr_scheduler_type` | `cosine` |
| `model_max_length` | 1024 |

**Describe and Respond.**

```
User:   <image>\n Detail the composition and subjects within the frame.
Model:  <image description>
User:   <image>\nImage description:\n<image description>\n<question>
Model:  <response>
```

**Evaluation details.** We use the evaluation scripts provided by LLaVA (Liu et al., 2023a) for all evaluations. It is important to note that the potential new evaluation scripts and prompts used to report the results for LLaVA-v1.6 have not been released at the time of writing this manuscript. This may cause discrepancies in the evaluation results of the original model.

**Compute resources.** Experiments were conducted on NVIDIA RTX A6000 GPU clusters. The entire self-training process for LLaVA v1.5 (7B) and LLaVA v1.6 (7B), using $6k$ image data and $5k$ reused instruction fine-tuning data, takes approximately 6 hours on 4 GPUs. The evaluation process of STIC for the benchmarks typically varies from 2 to 8 hours, mainly depending on the test set size.

## C   Experimental Results

Table 7: Detailed performance of STIC compared with the original VLM model on the MM-Bench dev set.

| Model | MMBench | | | | | |
|---|---|---|---|---|---|---|
| | LR | AR | RR | FP-S | FP-C | CP |
| Original | 35.6 | 65.8 | 61.7 | 64.5 | 49.0 | 80.7 |
| w/ STIC | **42.4** | **69.3** | **67.8** | **68.6** | **60.8** | **83.1** |

Table 8: Detailed performance of STIC compared with the original VLM model on the MM-Vet benchmark.

| Model | MM-Vet | | | | | |
|---|---|---|---|---|---|---|
| | rec | ocr | know | gen | spat | math |
| Original | 43.1 | 40.6 | 29.6 | 32.5 | 44.7 | 15.4 |
| w/ STIC | **45.7** | **42.5** | **30.4** | **34.9** | **45.1** | **22.7** |

### C.1   Example Outputs

In Figure 12 and 13, we demonstrate more output examples of the original LLaVA-v1.6 and STIC.

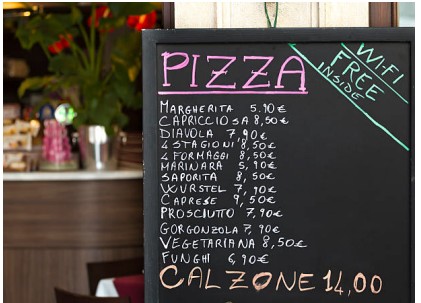

**Query**: I am getting one funghi pizza and one prosciutto pizza. How much should I pay in total?

**Base (LLaVA-v1.6 7B)**:
Based on the menu prices, the total cost for one funghi pizza and one prosciutto pizza **would be 14,00€.**

**STIC (LLaVA-v1.6 7B)**:
Based on the menu displayed in the image, **the price for a funghi pizza is 6,90€, and the price for a prosciutto pizza is 7,90€.** Therefore, the total cost for one funghi pizza and one prosciutto pizza **would be 14,80€.**

Figure 12: Response examples from original LLaVA-v1.6 and STIC (LLaVA-v1.6) in MM-Vet.

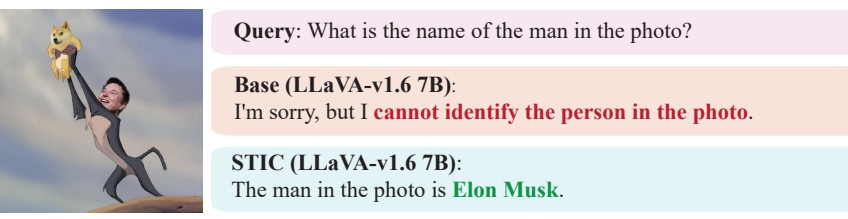

**Query**: What is the name of the man in the photo?

**Base (LLaVA-v1.6 7B)**:
I'm sorry, but I **cannot identify the person in the photo**.

**STIC (LLaVA-v1.6 7B)**:
The man in the photo is **Elon Musk**.

Figure 13: Response examples from original LLaVA-v1.6 and STIC (LLaVA-v1.6) in LLaVA-Bench.

## C.2 t-SNE Visualization Analysis

ScienceQA, TextVQA, MathVista, and ChartQA were chosen because there are at least $1,000$ images in the test set, providing enough data points for analysis.

**MSCOCO vs ScienceQA.** The gain achieved by STIC on ScienceQA was $6.4\%$, the highest across all seven benchmarks. As evident from Figure 14 (a), the images in ScienceQA have substantial overlap with those in MSCOCO. This suggests that the image comprehension capabilities developed by STIC through self-training on MSCOCO translated effectively to the scientific reasoning tasks in ScienceQA.

**MSCOCO vs TextVQA.** STIC yielded a gain of $4.9\%$ on TextVQA, one of the higher gains observed across the benchmarks. Figure 14 (b) shows a significant overlap between the image distributions of TextVQA and MSCOCO. This indicates that the enhanced image understanding from self-training on MSCOCO proved beneficial for the text-based visual question answering tasks in TextVQA.

**MSCOCO vs MathVista.** On MathVista, STIC achieved a gain of $2.4\%$, which, while still notable, was lower compared to other benchmarks. As Figure Figure 14 (c) illustrates, the images in MathVista have limited overlap with those in MSCOCO. Moreover, MathVista features diverse image types and mathematical reasoning tasks that pose additional challenges beyond image comprehension. These factors likely contributed to the more modest performance gain.

**MSCOCO vs ChartQA.** STIC attained a high gain of $5.1\%$ on ChartQA, despite the images in ChartQA having minimal overlap with those in MSCOCO, as shown in Figure 14 (d). This seemingly contradictory result can be explained by considering that the improved image comprehension capabilities developed by STIC play a fundamental role in understanding and reasoning about the charts in ChartQA. Thus, even with limited distributional similarity, the enhanced perception skills proved valuable for this benchmark.

## C.3 Discussion with POVID.

We detail the differences between STIC and POVID. In POVID, the dispreferred response is generated either by adding Gaussian noise to the original image or by manually injecting hallucinations into the ground truth completion, using the labeled object information of the images. In contrast, STIC (1) specifically targets the image description task, (2) constructs preference datasets exclusively from *unlabeled* images using self-generated content for both preferred and dispreferred responses, (3) employs an automatic model generation process without manual injections or modifications, and (4) utilizes only a small portion of SFT data for *instruction-following fine-tuning* with uniquely infused model descriptions. Lastly, we compare the data types and scales used in POVID and STIC in Figure 15.

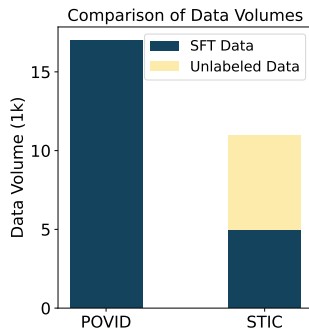

Figure 15: Data comparison.

## C.4 Investigation of Prompt Quality

Table 9 presents an additional experiments using DaR to demonstrate prompt quality. We compared normal prompts from our main paper (e.g., "Illustrate the details of the picture.") with the hallucination prompts and well-curated prompts used for DPO pair generation. The results show an expected discrepancy in QA performance: hallucination prompts significantly decreased performance, while well-curated prompts maintained a decent performance. We also included results based on a prompt proposed by Llama-3 8B and filtered using the same restrictions. The performance difference between GPT-4 and Llama-3 8B prompts underscores the quality of GPT-4's proposals.

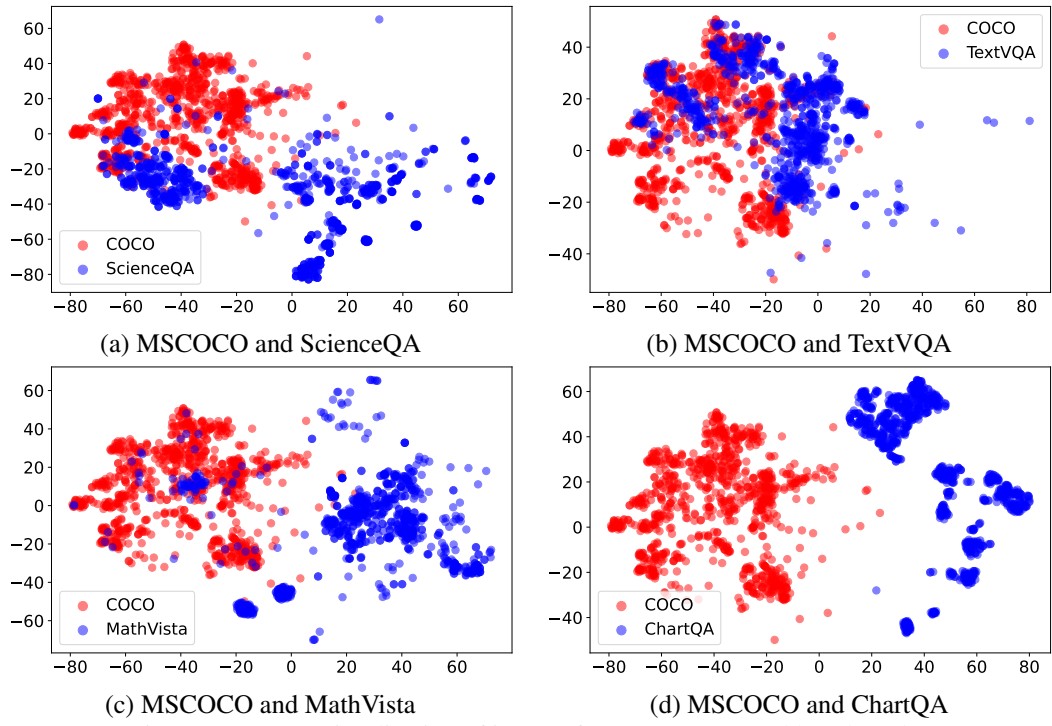

(a) MSCOCO and ScienceQA  (b) MSCOCO and TextVQA

(c) MSCOCO and MathVista  (d) MSCOCO and ChartQA

Figure 14: t-SNE visualization of images from MSCOCO and benchmarks.

Table 9: Test performance of `llava-v1.6-mistral-7b` using various prompts with DaR. We evaluate prompt quality using DaR as a prompting method. DaR=`None` represents the original LVLM model's performance. Normal prompt refers to the simple prompt we used for DaR in our paper. GPT-4's well-curated prompt refers to the prompt we used for preferred response generation, and we include Mistral 7B's curated prompt for additional comparison.

| Model | DaR | LLaVA-Bench | MM-Vet | MMBench |
|---|---|---|---|---|
| | None | 77.3 | 42.2 | 63.7 |
| | Normal Prompt | 78.5 $_{(+1.2)}$ | 42.3 $_{(+0.1)}$ | 50.7 $_{(-13.0)}$ |
| LLaVA-v1.6 (7B) | Hallu Prompt | 73.7 $_{(-3.6)}$ | 40.5 $_{(-1.7)}$ | 40.7 $_{(-23.0)}$ |
| | Well-curated (Llama-3 8B) | 77.2 $_{(+0.1)}$ | 40.0 $_{(-2.2)}$ | 60.1 $_{(-3.6)}$ |
| | Well-curated (GPT-4) | 79.1 $_{(+2.1)}$ | 42.9 $_{(+0.7)}$ | 60.9 $_{(-2.8)}$ |

# D   Limitations and Future Work

While STIC demonstrates significant performance gains across a diverse set of benchmarks, there are still some limitations to be addressed in future work. First, STIC achieves a relatively small accuracy gain on MathVista compared to other benchmarks. This is likely because MathVista features various mathematical reasoning abilities across a wide range of image types, from elementary school to college level problems, which go beyond the scope of image comprehension. In contrast, STIC uses MSCOCO, containing only natural images, for constructing its preference data. To further improve performance on tasks like MathVista, future work could expand the source image types and generate task-specific image description data that better aligns with the target benchmark.

Second, while our approach of using good prompts for positive examples and bad prompts or corrupted images for negative examples is straightforward and effective, this preference data may be insufficient for scenarios requiring nuanced understanding of image details. More sophisticated strategies for preference data construction could potentially further boost fine-tuning performance with the DPO loss.

Additionally, our current method relies on a two-stage process of first training on the preference data and then performing description-infused fine-tuning. An interesting direction for future work would be to explore integrating these stages into a single end-to-end training process, which could potentially lead to even greater synergies and performance gains.

Lastly, while we have demonstrated the scalability of STIC by doubling the amount of preference data, leading to further improvements, we have not yet explored the upper limits of this scaling. It is

possible that even larger amounts of self-training data could lead to diminishing returns at some point. Characterizing the scaling behavior of STIC more fully is an important direction for future research.

Despite these limitations, we believe STIC represents an important step towards leveraging the vast amounts of unlabeled image data available to enhance the image comprehension capabilities of large vision-language models in a cost-effective manner.

# E    Broader Impacts

The development of STIC, our self-training approach for enhancing the image comprehension capabilities of large vision language models (LVLMs), presents several potential societal impacts. Positively, our method can democratize access to advanced vision-language models by significantly reducing the cost and effort required for fine-tuning, making state-of-the-art LVLMs more accessible to researchers and organizations with limited resources. This can accelerate advancements in healthcare, education, and environmental monitoring, where improved image comprehension can lead to better diagnostic tools, personalized learning experiences, and more effective environmental protection measures. Additionally, by encouraging the reuse and recycling of existing data, STIC aligns with sustainable AI practices, promoting efficient use of computational and data resources.

However, there are potential negative societal impacts that must be considered. Enhanced LVLM capabilities could be misused for generating disinformation, creating fake profiles, or conducting unauthorized surveillance, contributing to the spread of misinformation and erosion of public trust. Fairness considerations are crucial, as biased training data may lead to outputs that disproportionately impact specific groups, resulting in unfair treatment or discrimination. Privacy concerns also arise from using self-generated data, particularly if models are trained on sensitive or personal visual content without proper consent. To mitigate these risks, strategies such as gated release of models, robust fairness audits, diverse data inclusion, and enhanced transparency about the technology's limitations and risks are essential to ensure responsible use.

