# OpenReview forum: "Enhancing Large Vision Language Models with Self-Training on Image Comprehension"
_NeurIPS.cc/2024/Conference — NeurIPS 2024 poster_

### Official Review · Reviewer_uhYN · 2024-07-02

**Soundness:** 3
**Presentation:** 3
**Contribution:** 2
**Rating:** 6
**Confidence:** 3

**Summary:**

This paper addresses the problem of acquiring high-quality fine-tuning data for large vision language models (LVLMs) with minimum human effort. The paper presents STIC (Self-Training on Image Comprehension). STIC contains two stages: image comprehension self-training and description-infused fine-tuning. LVLM first generates descriptions based on images and corrupted images and regards each sample as preferred and dis-preferred responses, respectively. The base LVLM is trained on the generated preference data leveraging the directed preference optimization (DPO) framework. Next, the LVLM is fine-tuned on instruction following data. Experimental results show that STIC outperforms the compared methods on diverse benchmarks.

**Strengths:**

- S1: Overall, the manuscript is well-written and easy to read. Preliminaries and figures help readers understand the paper.
- S2: The paper presents diverse analysis and discussion in experiments.

**Weaknesses:**

- W1: The manuscript has limited soundness for several reasons. For example, the authors did not validate the effectiveness of description-infused fine-tuning. In Table 2, much of the performance gain comes from the prompting method (DaR) which is outside STIC.
- W2: While there is a rich literature on existing self-training algorithms [1,2,3,4,5], the paper only discusses recent self-improvement systems, especially in the context of LLMs. This limited investigation leads readers to question what the technical contribution of STIC is compared with the existing line of semi-supervised learning (or self-training) research.
- W3: The paper experimented with a fairly small amount of unlabeled images (6k and 12k images). What and how much data should be used to maximize the performance has not been thoroughly investigated.


**References**

[1] Self-training with noisy student improves imagenet classification. Xie et al., CVPR 2020.

[2] Rethinking pre-training and self-training. Zoph et al., NeurIPS 2020.

[3] Fixmatch: Simplifying semi-supervised learning with consistency and confidence. Sohn et al., NeurIPS 2020.

[4] Revisiting self-training for neural sequence generation. He et al., ICLR 2020.

[5] The dialog must go on: improving visual dialog via generative self-training. Kang et al., CVPR 2023.

**Questions:**

- Q1: In Algorithm 2, why aren’t the model-generated description $\mathbf{y}_{\mathrm{des}}$ used for fine-tuning?
- Q2: What is the motivation of the stage 2 (Description-infused fine-tuning)? It is not directly related to the motivation of the paper (difficulty of obtaining high-quality fine-tuning data).
- Q3: How can we validate the effectiveness of description-infused fine-tuning? Did the authors check the performance of the method which did not use the model description?
- Q4: Table 1 just shows the performance with and without STIC. How much does each stage contribute to performance improvement?
- Q5: Table 2 shows that the describe-and-respond (DaR) prompting method improves overall performance. DaR was not mentioned until the section for experiments. Why did not the authors describe DaR in detail in the method section?
- Q6: Is there any reason why STIC randomly selects unlabeled images? Some self-training algorithms present the methods for selecting unlabeled data to learn.
- Q7: How can we guarantee that all samples generated by the LVLM inputting clean images are preferred responses?

**Typos**

Figure 1: SPIC to STIC

**Limitations:**

The limitations were adequately addressed.

---

> ### Author Rebuttal · Authors · 2024-08-07
>
> Thank you for the detailed feedback. Please find our responses below. We hope that our clarifications and additional experiments resolve the misunderstanding.
>
> ### **W1/Q3: In Table 2, much of the performance gain comes from DaR.**
>
> We respectfully point out that there is a misunderstanding of the results presented in Table 2. As we have discussed in line 278-283 of our submission, Table 2 (row 1-2) showed that DaR prompting alone even results in degradations, while only combining with STIC achieves the best result. And contrary to the your conclusion, the 3rd row for **STIC without DaR** clearly showed that STIC fine-tuning itself leads to significant improvements across all tasks, with the average increasing from 54.8 to 57.6. Notably, it achieves the best results on MMBench and MM-Vet, proving its soundness. In the 4th row, we combined STIC with DaR. While DaR prompting alone is not sufficient, the integration of STIC significantly boosts the model's image comprehension ability and consequently effectiveness of DaR.
>
> More analysis of the stages is provided in [our global rebuttal (Global A2)](https://openreview.net/forum?id=FZW7Ctyjm3&noteId=VV3MK6FiS7).
>
> ### **W2: Existing self-training methods for vision models are not discussed.**
>
> Thank you for mentioning the references, we will include them in discussion in our revision. Specifically:
> - **Similarity**: The major goal shared by the mentioned papers and STIC is designing an effective way of leveraging unlabeled data to improve the performance of current models.
> - **Differences**:
>     1. (Focus/Model) The mentioned papers focus on representation learning of deep learning models. Meanwhile, STIC focuses on the vision LLMs, and the backbone remains an LLM. While previous deep learning models are grounded in representation and further specialized in specific task, LLMs are autoregressive and easily generalize to different tasks. Instead of training for better image representations, STIC aims to gather synthetic data for the LLM to produce higher-quality responses to a query on an image.
>     2. (Algorithm) We focus on **alignment fine-tuning**. Notably, classic self-training algorithms for vision models do not employ alignment algorithms like RLHF or DPO. While providing a positive training signal is beneficial, having negative examples is crucial for the success of LLM alignment. As shown in Table 3 of our submission, including only positive examples for SFT is not as effective as having pairwise preference data.
>
> ### **W3: A small amount of unlabeled images**
>
> In our discussion with POVID in Section 6 Figure 5, we highlighted the **data efficiency** of STIC. While POVID uses 17k SFT data, STIC achieves better results with a total of 11k data (5k SFT and 6k unlabeled). STIC requires a smaller amount of data to achieve significant improvements.
>
> We conducted an additional experiment using 30k unlabeled images, as shown in Figure 1 of our [attached one-page pdf](https://openreview.net/attachment?id=VV3MK6FiS7&name=pdf). The results demonstrate the scalability of STIC when applied to larger datasets.
>
> ### **Q1: Why aren’t model-generated description 𝑦_des used for fine-tuning?**
>
> We will correct this typo in Algorithm 2 in our revision. The correct data used for fine-tuning should be $([v^{(i)}, y_{des}, x^{(i)}], y^{(i)})$. Algorithm 2 indeed aims to use the model-generated description and append it before the question prompt for fine-tuning.
>
> ### **Q2: Motivation of Stage 2**
>
> As explained in our method section, stage 2 is aimed to further fine-tune the model to leverage self-generated image descriptions for downstream tasks, and thus help ground its reasoning ability on such descriptions. While stage 1 improves the model’s ability in image description, it does not focus on leveraging these descriptions for subsequent tasks such as question answering. Stage 2 fine-tunes the model by reusing a small amount of its SFT data and infusing it with self-generated descriptions to specifically strengthen model's reasoning ability based on descriptions.
>
> ### **Q4: How much does each stage contribute to performance improvement?**
>
> Thank you for raising this important point. Please see our response in [Global A2](https://openreview.net/forum?id=FZW7Ctyjm3&noteId=VV3MK6FiS7).
>
> ### **Q5: DaR was not mentioned early.**
>
> DaR was discussed in line 211-213 of our submission and was not elaborated due to page limit. Please see our elaborated explanation in [Global A3 of our global rebuttal]((https://openreview.net/forum?id=FZW7Ctyjm3&noteId=VV3MK6FiS7)). We will add a paragraph in our method section in revision.
>
> ### **Q6: Why STIC randomly selects unlabeled images.**
>
> Data selection is indeed a promising future direction and we will include this discussion in revision.
>
> The current implementation of STIC randomly selects unlabeled images as it simplifies the process and ensures a broad and diverse sampling of data. In Table 3 of our [attached one-page pdf](https://openreview.net/attachment?id=VV3MK6FiS7&name=pdf), we included an experiment using the same amount of images but more diverse distribution (Vision Flan) for stage 1. Notably, the increased diversity led to further improvements in STIC, suggesting the potential for enhancement with better sets of unlabeled images.
>
> ### **Q7: How can we guarantee that all samples generated by LVLM on clean images are preferred?**
>
> Regarding the specific question on clean images vs. corrupted images, we note that this approach has become widely-used and tested in concurrent works focusing on LVLM alignment [1]. In our paper, Figures 3, 9, and 10 also showed that image corruptions indeed cause observable declines in model output quality. Please also see our response in the [global rebuttal (Global A1)](https://openreview.net/forum?id=FZW7Ctyjm3&noteId=VV3MK6FiS7) for a detailed explanation on preference alignment.
>
> [1] Aligning modalities in vision large language models via preference fine-tuning.

---

> > ### Author Response · Authors · 2024-08-11
> > **Invitation for discussion**
> >
> > Dear reviewer uhYN,
> >
> > Thank you again for your detailed and valuable feedback to this paper. We hope that our responses and clarifications have adequately addressed your questions and concerns. Specifically,
> >
> > 1. We provided detailed clarifications on the results of Table 2 and explanations on DaR (Global A3).
> > 2. We added a discussion paragraph on the similarities and differences with the mentioned related works, emphasizing our specific focus. These works will be incorporated into our revision.
> > 3. We conducted additional experiments scaling up the data to 30k and explained the data efficiency of our method.
> > 4. We provided clarifications and explanations for each of your specific questions.
> >
> > We hope these responses adequately address your concerns. If you have any further questions about our rebuttal, we'd be happy to provide additional information or clarification. Thank you once again for your time and efforts!

---

> > > ### Comment · Reviewer_uhYN · 2024-08-13
> > > **Thanks for the response**
> > >
> > > I thank the authors for providing detailed responses to my concerns and questions. I read the responses from the authors and the other reviewers' comments as well, and most of my concerns are addressed. I will raise my initial rating accordingly. Thanks again!

---

> > > > ### Author Response · Authors · 2024-08-13
> > > >
> > > > Thank you for replying and providing encouraging feedback on our rebuttal! We are glad that we addressed most of the concerns.

---

### Official Review · Reviewer_s8oQ · 2024-07-09

**Soundness:** 3
**Presentation:** 3
**Contribution:** 3
**Rating:** 4
**Confidence:** 3

**Summary:**

Summary:
The paper presents STIC, a method to enhance LVLM by reducing the need for labeled data. STIC generates image descriptions using unlabeled images and improves reasoning by reusing existing instruction-tuning data. It demonstrates performance gain across seven benchmarks, showing potential to effectively leverage vast quantities of unlabeled images for self-training.

**Strengths:**

Strength:
1. The proposed method improves VLLM's performances with efficient cost during data collection.
2. The performance improvement is consistent on diverse benchmarks and achieves an average accuracy gain of 3.8%, which is quite significant.

**Weaknesses:**

Weakness:
1. Interesting idea about using different prompts to generate both good and bad captions. However, I wonder if there is any method needed to ensure the correctness of the prompts generated by the "step-by-step" prompt strategies. Based on my own observations, VLLM is not good at following instructions, which means the description will be even worse if the prompts are complicated. Have the authors observed similar problems?

**Questions:**

Have the authors changed the default latex template? There should be no anonymous submission ID. And it seems that the authors gain more space because they remove the anonymous authors part.

**Limitations:**

Yes

---

> ### Author Rebuttal · Authors · 2024-08-07
>
> We appreciate your feedback and make clarifications as below.
>
> ### **W1. If there is any method needed to ensure the correctness of the well-crafted prompt?**
>
> Thank you for raising this important question. The concern regarding the complexity of prompts is indeed crucial. To address this, we implemented restrictions and human filtering on multiple candidate prompts generated by GPT-4 to ensure they function as intended. This process can be viewed as a behavior distillation from a stronger model.
>
> To ensure effectiveness, we tested these prompts on MSCOCO samples and verified the LVLM’s instruction-following quality through human evaluation. We provide a detailed explanation along with additional experiments in our [global rebuttal (Global A1)](https://openreview.net/forum?id=FZW7Ctyjm3&noteId=VV3MK6FiS7), and we will include this discussion and the additional experiments in our revision.
>
> Regarding the concern about instruction-following abilities, which may be weaker in untrained models, we found that DPO alignment fine-tuning significantly enhances this capability. This allows the model to learn not only the preferred response but also to identify and avoid dispreferred and often erroneous responses.
>
> Verification remains an intrinsically challenging problem, especially since there is no ground truth answer in the context of image description tasks with unlabeled images. To mitigate this, we applied a human filtering strategy for the prompts, which has proven effective and not costly. To scale up our method and move towards a fully autonomous framework, we plan to involve a critic model in the process in our future studies.
>
> ### **Q1. Have the authors changed the default latex template?**
>
> We apologize for any confusion. We did not intentionally alter the template. It appears we inadvertently included the submission ID. We appreciate your understanding and will ensure this mistake is corrected. We checked that with the new template, the paper can still be fit into the page limit.

---

> > ### Author Response · Authors · 2024-08-11
> > **Invitation for discussion**
> >
> > Dear reviewer s8oQ,
> >
> > Thank you again for your time and feedback to this paper. We hope that our clarifications have adequately addressed your questions. Specifically, we provided detailed explanations into the prompts of our method and further discussion in Global A1. We are following up to inquire if there are any remaining questions. We are more than happy to further discuss and provide clarifications.
> >
> > Thank you once again for your time and efforts!

---

> > > ### Author Response · Authors · 2024-08-13
> > >
> > > Dear reviewer s8oQ,
> > >
> > > Thank you again for taking the time to review our paper. We appreciate your acknowledgment of our work's soundness, presentation, and contribution.
> > >
> > > In response to the feedback received, we have included extensive additional experiments as well as detailed clarifications in our rebuttal. While other reviewers have responded positively to our rebuttal, we hope to adequately address your concern as well. We would greatly appreciate your attention to our specific response regarding your question on prompt design, as we believe it addresses your primary concern. It would be great to also give us the opportunity to provide more details and further improve our work. Thank you!

---

### Official Review · Reviewer_B7oX · 2024-07-14

**Soundness:** 2
**Presentation:** 3
**Contribution:** 3
**Rating:** 6
**Confidence:** 4

**Summary:**

This paper proposes a two-stage method to enhance Large Vision Language Models (LVLMs) using unlabeled images. In the first stage, well-designed good and bad prompts are used to make the LVLM generate preferred and dis-preferred completions, respectively, conditioned on the unlabeled images (from COCO). Then, direct preference optimization (DPO) is used to fine-tune the LVLM using the generated preferred and dis-preferred completions. In the second stage, the fine-tuned model is used to generate descriptions for images in an instruction-tuning dataset (from LLaVA's data). The generated descriptions are inserted into the instruction-tuning data to fine-tune the model. The fine-tuned model shows significant improvement over the baseline (the LVLM before fine-tuning) on seven VLM benchmarks.

**Strengths:**

- The biggest strength is the significant improvement over the baseline LVLM achieved by the proposed method. There is an average improvement of 4 points on the seven VLM benchmarks.
- The proposed method mainly leverages unlabeled images for training, which gives the proposed method a great potential to use a vast amount of unlabeled images. The authors also show using more unlabeled images for training improves the performance.

**Weaknesses:**

- The paper raises some questions unanswered.
  - The effect of the prompt set is less explored in the paper. It seems that the prompts play a crucial role in data construction. It is unclear how the authors designed the well-curated captioning prompt and the hallucination prompt set. Are there any principles behind the design? Especially for the well-curated captioning prompt that generates the preferred data, how do different design choices affect the final model performance in the evaluation?
  - Is the performance gain dependent on the MSCOCO data set? Do other image datasets (such as Flickr30k) work in stage 1?
- It seems counterintuitive that fine-tuning an LVLM on MSCOCO will help improve its performance on science-related benchmarks like ScienceQA. It will help us better understand the mechanism by comparing the model generations in the benchmarks before and after using the proposed method.
- The description of describe-and-respond (DaR) prompting is a bit unclear. I could not fully understand the setting.

**Questions:**

- What is the difference between the image captioning prompt set $P$ and the well-curated captioning prompt (Algorithm 1)?
- What is the prompt $x$ used in DPO in stage 1?
- What is the model performance if only stage 1 is performed (i.e., without stage 2's SFT)?

**Limitations:**

See Weaknesses.

---

> ### Author Rebuttal · Authors · 2024-08-07
>
> Thank you very much for your support and the constructive feedback that helped us improve our work. Please see our detailed response with additional experiments below.
>
> ### **W1a: Principles behind the prompt design.**
>
> In short, we use GPT-4 to generate and sample multiple initial prompts, which are then refined through human filtering. To ensure effectiveness, we test these prompts on MSCOCO samples, verifying their ability to produce well-structured and relevant responses from the model. Using DaR performance as an evaluation to the prompts (Table 2 of the [attached one-page pdf](https://openreview.net/attachment?id=VV3MK6FiS7&name=pdf)), we showed that the better crafted prompts result in better performance for DaR even on plain models.
>
> Please see our detailed response in [Global A1 in the global rebuttal](https://openreview.net/forum?id=FZW7Ctyjm3&noteId=VV3MK6FiS7). We will include the discussion and experiments in our revision.
>
> ### **W1b: Is the performance gain dependent on MSCOCO data?**
>
> Thank you for raising this important point. To address this, we conducted additional experiments using images from various sources. Specifically, we utilized the Vision Flan dataset (VFLAN: https://huggingface.co/datasets/Vision-Flan/vision-flan_191-task_1k) for stage 1 image comprehension self-training. This dataset includes images from 191 diverse vision tasks, providing a broad spectrum of image types.
>
> We ensured a fair comparison by maintaining the same sample size (randomly sampled 6,000 images) and have presented the experimental results in Table 3 of the [attached one-page pdf](https://openreview.net/attachment?id=VV3MK6FiS7&name=pdf). The results indicate that our approach improves consistently across different datasets, demonstrating its robustness and adaptability. Notably, the increased diversity of VFLAN led to further improvements in STIC, suggesting the potential for even greater enhancement with better sets of unlabeled images. This finding aligns with our analysis in Figure 8 of the main paper, where we observed a positive correlation between the overlap of MSCOCO's image distribution with a benchmark and the performance gains achieved by STIC on that benchmark.
>
> ### **W2: It’s better to compare model generations before and after using STIC.**
>
> Thank you very much for the suggestion. In Figure 1 of our submission, we showed an example of model generation before and after using STIC. In line 75-76, we provided a discussion that STIC improves the model response by successfully identifying the key visual information for subsequent reasoning.
>
> In the [attached one-page pdf](https://openreview.net/attachment?id=VV3MK6FiS7&name=pdf), we included two additional model generation examples before and after applying STIC, as illustrated in Figures 2 and 3. Despite the task being focused on mathematical reasoning, STIC enhanced the model’s response by improving its image comprehension capabilities. While the original model merely identified one of the recognized numbers in the image as the final answer, the STIC fine-tuned model was able to interpret the meaning of each number, describe them accurately, and perform reasoning based on this understanding.
>
> Furthermore, in Figure 8 and its corresponding ablation study of our main paper, we examined the improvement in ScienceQA, where it shares a great overlap between the MSCOCO image distribution.
>
> ### **W3: Further explanations on DaR prompting.**
>
> We apologize for the lack of clarity in our initial presentation on DaR. Please see our explanation in [Global A3 of our global rebuttal](https://openreview.net/forum?id=FZW7Ctyjm3&noteId=VV3MK6FiS7). We will add a paragraph in our method section to fully illustrate DaR in the revision.
>
> ### **Q1: Difference between the image captioning prompt set and the well-curated prompt.**
>
> We detailed the image captioning prompt set in our answer to Q6. Here are the key differences between the image-captioning prompt and well-curated captioning prompt:
>
> **Image Captioning Prompt Set**: This set comprises concise and straightforward prompts designed to elicit basic image descriptions. These prompts typically ask the model to describe the image in simple terms without additional guidance or structure.
>
> - Purpose: The prompts in set P serve as the target task.
>
> **Well-Curated Prompt**: These prompts are designed to be more elaborate and structured, crafted to elicit higher-quality responses by encouraging the model to engage in a more systematic reasoning process.
>
> - Purpose: Generate superior responses to provide a learning signal for preferred/positive responses.
> This process alone (without the negative/dispreferred responses) is similar to the currently popular method called system 2 distillation [1], where the model is fine-tuned on its responses generated from a more complex, step-by-step prompt. The goal is to teach the model to apply the enhanced reasoning patterns induced by the well-curated prompts when responding to the simpler prompts (e.g. in set P).
>
> We will add the above explanation in our next revision.
>
> [1] Distilling System 2 into System 1
>
> ### **Q2: What is the prompt 𝑥 used in DPO in stage 1?**
>
> We included some of the prompts below due to character limit and will include the full set of eight prompts in our future revision.
>
> - "Illustrate the details of the picture.",
> - "Summarize the visual content presented.",
> - "Explain what is depicted in the photograph.",
> - …
>
> ### **Q3: Model performance if only stage 1 is performed.**
>
> Thank you for raising this important point. In short, while stage 1 focuses exclusively on enhancing the perception capabilities of LVLM, it still notably improves performance on downstream VQA tasks (1.1\% accuracy gain on ScienceQA).
>
> Please see our detailed response in [Global A2](https://openreview.net/forum?id=FZW7Ctyjm3&noteId=VV3MK6FiS7) on the progression of stages.

---

> > ### Author Response · Authors · 2024-08-11
> > **Invitation for discussion**
> >
> > Dear reviewer B7oX,
> >
> > Thank you again for your support and constructive comments. We hope that our responses and clarifications have adequately addressed your questions and concerns.
> >
> > Specifically, we provided detailed explanations toward the prompt design (Global A1) and DaR (Global A3). We further added an experiment on unlabeled images from a different distribution than MSCOCO, where we observed that a more diverse unlabeled image data can provide better improvement for STIC. Regarding W2 and Q3, we provided specific generation examples before and after STIC, as well as its stage-wise performance on ScienceQA.
> >
> > We would like to inquire if there are any questions about our rebuttal, for which we're happy to provide additional information and further clarifications. Thank you once again for your time and efforts on this paper!

---

> > ### Comment · Reviewer_B7oX · 2024-08-13
> >
> > Thank you for the detailed responses. My questions are addressed. I will raise my rating to 6.

---

> > > ### Author Response · Authors · 2024-08-13
> > >
> > > Thank you for replying! We greatly appreciate your positive feedback on our rebuttal.

---

### Official Review · Reviewer_qFQt · 2024-07-15

**Soundness:** 3
**Presentation:** 3
**Contribution:** 3
**Rating:** 6
**Confidence:** 3

**Summary:**

This paper introduces Self-Training on Image Comprehension (STIC), which emphasizes a self-training approach specifically for image comprehension. First, the model self-constructs a preference dataset for image descriptions using unlabeled images. Preferred responses are generated through a step-by-step prompt, while dis-preferred responses are generated from either corrupted images or misleading prompts. To further self-improve reasoning on the extracted visual information, the model reuses a small portion of existing instruction-tuning data and appends its self-generated image descriptions to the prompts. Improvements in several benchmarks are reported.

**Strengths:**

1. The challenges of self-training with VLM are discussed, which is appreciated.

2. The proposed STIC approach is claimed to be a novel two-stage self-training method that targets both image perception and reasoning over images and texts, which is intriguing.

3. STIC does not require pre-labeled information on the images,

4. The methodology of constructing dis-preferred data using bad prompting is pretty interesting.

**Weaknesses:**

1. The experiments are conducted with 7B-level LLava 1.5 and 1.6. The method's scalability remains questionable.

**Questions:**

What if we try the proposed method with smaller or bigger models or other ViT families (e.g., EVA-CLIP models)? With a higher representational capacity, will the model benefit more or less from self-training?

**Limitations:**

The authors adequately addressed the limitations.

---

> ### Author Rebuttal · Authors · 2024-08-07
>
> Thank you for your strong support and valuable feedback! We address your major comment as follows.
>
> ### **W1: The scalability of STIC**
>
> To explore STIC's applicability to models with higher representation capacity, we conducted supplementary experiments using LLaVA-v1.6 (Vicuna-13B).
>
> | **Model**        | **LLaVA-Bench** (Conv) | **LLaVA-Bench** (All) | **MM-Vet** (Gen) | **MM-Vet** (All) | **MMBench** |
> |------------------|-----------------|-----------------|-----------------|------------|-------------|
> | LLaVA-v1.6 (7B)  | 61.3 | 77.3         | 32.5   | 42.2       | 63.7        |
> | LLaVA-v1.6 (13B) | 73.8 | 84.5       | 45.2     | 48.9       | 70.6        |
> | LLaVA-v1.6 (13B) w/ STIC         | **78.1** | **85.6**     | **49.4**       | **50.5**       | **72.3**        |
>
> Table 1 in the [attached one-page pdf](https://openreview.net/attachment?id=VV3MK6FiS7&name=pdf) shows the detailed and comprehensive experiment results. Due to compute and time constraints, we included the three benchmarks (LLaVA-Bench, MM-Vet and MMBench) that include various different tasks and can comprehensively evaluate the model’s performance. We used the same images for STIC fine-tuning as in our experiments for LLaVA-v1.6 (Mistral-7B) to ensure fairness and the same set of hyperparameters due to time constraint. The improvements observed with LLaVA-v1.6 (Vicuna-13B) demonstrate that STIC is not only effective with smaller models but also scales well with larger or more capable LVLMs. It also shows potential for further improvement through hyperparameter tuning, data filtering, and enhanced data generation.
>
> We hope that our additional experiments address your raised concerns. Let us know if there remain further questions, and we are happy to discuss them.
>
> ### **Q1: What if we try the proposed method with different LVLM models?**
>
> Thank you for this insightful question. In our original and additional experiments, we employed STIC with LLaVA-v1.5 and LLaVA-v1.6, incorporating various LLM backbones at different scales, specifically Vicuna-7B, Mistral-7B, and Vicuna-13B. These models, all incorporating strong visual encoders from the CLIP family, demonstrated effective improvements with STIC. While our current exploration of different models was constrained by time and computational resources, we recognize the importance and potential of exploring a wider range of LVLM models. Future work could investigate models with diverse architectures and LLM backbones, such as Llama-3, to further explore the potential of our proposed method.

---

> > ### Author Response · Authors · 2024-08-11
> > **Invitation for discussion**
> >
> > Dear reviewer qFQt,
> >
> > Thank you again for your strong support and constructive feedback. We hope that our responses have adequately addressed your questions. Specifically, we added the additional experiment on scaling up model sizes for STIC and provided discussion on further extending it to various models. We would like to inquire if you have further questions regarding our rebuttal. We are more than happy to discuss any remaining questions and provide additional details.
> >
> > Thank you once again for your time and efforts on this paper!

---

> ### Comment · Reviewer_qFQt · 2024-08-14
> **After rebuttal**
>
> I keep my original score and tend to accept this paper.

---

### Author Rebuttal · Authors · 2024-08-07

We sincerely thank all the reviewers for their insightful and encouraging feedback on our manuscript. We are grateful for the recognition of the significant performance gains achieved by STIC (B7oX, s8oQ), the novelty and efficiency of STIC (qFQt, s8oQ), the effective use of unlabeled images (qFQt, B7oX) and the comprehensive analysis and clarity of our manuscript (uhYN).

In response to the comments, we provided additional experiments in the [attached one-page pdf](https://openreview.net/attachment?id=VV3MK6FiS7&name=pdf). Specifically, the results include

1. **Scaling up to 13B model (Table 1)**: STIC is effective in improving larger-scale models as well, with further improvement potential in hyperparameter tuning, data filtering and data scaling.

2. **Quantitative analysis of the prompt quality (Table 2)**: the prompts with better quality (our well-crafted prompt derived from GPT-4) provides better performance of DaR on the original model.

3. **Effect of enhanced image diversity in unlabeled images (Table 3**): a more diverse unlabeled image data can provide better improvement for STIC, which aligns with our ablation study on image distributions.

4. **Effect of more unlabeled data from MSCOCO used in stage 1 (Figure 1)**: STIC scales well to larger datasets.

5. **More qualitative examples (Figures 2 and 3)** showing how STIC helped improve model performance even in mathematical VQAs.

### **Global A1: Explanation into our prompt design and data generation for alignment fine-tuning.**

Our prompt design for the well-crafted prompt focuses on quality and diversity. We use GPT-4 to generate and sample multiple initial prompts, which are then refined through human filtering. To ensure effectiveness, we test these prompts on MSCOCO samples, verifying their ability to produce well-structured and relevant responses from the model. The restrictions we apply to reflect the quality of the prompt include length (prompt must be between 60 to 150 words to balance informativeness and conciseness), diversity (prompt includes at least 3 distinct aspects or questions about the image to encourage comprehensive analysis), and specificity (while being general, the prompt contains at least 2-3 specific cues or keywords that can be adapted to various image contents).

More generally, instead of relying on explicit human labeling for each model generation pair as in RLHF, which can be very expensive, we adopt an "implicit" preference approach. We work under the assumption that responses generated from prompts that have differences in human preference lead to responses of the same preference with high probability. This approach allows us to create effective training data without the need for extensive human annotation.

Our goal is thus not to identify the best or worst prompt for the task, but rather to explore the differences between them. For the design of a good prompt, we aim to guide the model to provide a comprehensive and precise image description. The bad prompts are designed to elicit inaccurate descriptions by setting up a slightly different task (describe objects that would logically exist in the image) for the model. The key is that the discrepancy between good and bad prompts should result in pairs of responses that share the same implicit preference with high probability, which is sufficient for effective DPO training.

Table 2 in our attached PDF presents additional experiments using DaR to demonstrate prompt quality. We compared normal prompts from our main paper (e.g., "Illustrate the details of the picture.") with the hallucination prompts and well-curated prompts used for DPO pair generation. The results show an expected discrepancy in QA performance: hallucination prompts significantly decreased performance, while well-curated prompts maintained a decent performance. We also included results based on a prompt proposed by Llama-3 8B and filtered using the same restrictions. The performance difference between GPT-4 and Llama-3 8B prompts underscores the quality of GPT-4's proposals.

### **Global A2: Progression of stages.**

In the table below, we illustrate the sequential improvement in performance of STIC on ScienceQA. While stage 1 focuses exclusively on enhancing the perception capabilities of the LVLM, it still notably improves performance on downstream VQA tasks. Building on the improved image comprehension achieved in stage 1, stage 2 introduces an enhanced reasoning process that utilizes the model’s self-generated image descriptions and results in an even more significant gain. This enhancement further enables the model to self-augment its prompts with DaR, resulting in the substantial performance gains of 6.4% observed.

| Original | After Stage 1 | After Stage 2 | After Stage 2 with DaR |
| :---: | :---: | :---: |  :---: |
| 68.86 | 69.96 | 72.48 | 75.26 |

### **Global A3: Explanation on DaR.**

In line 211-213 and 275-284 of of our submission, we discussed DaR. Here, we provide further explanations. We proposed DaR as an additional and optional step that can be employed during inference time. Instead of directly obtaining the model's response to a particular question, DaR first prompts the model to describe the image, then appends this description to the question to finally obtain the answer:

User: `<image>\nDetail the composition and subjects within the frame.`

Model: `<image description>`

User: `<image>\nImage description:\n<image description>\n<question>`

Mode: `<response>`

This two-step approach helps the model to better contextualize the question by grounding its response in a detailed understanding of the image. **However, as shown in Table 2 of our submission, DaR alone does not notably improve the performance of a plain model. Instead, it shows the most substantial improvement when combined with STIC fine-tuning.** The foundational improvements made by STIC on the model’s image comprehension ability consequently improved the effectiveness of DaR.

---

### Decision · Program_Chairs · 2024-09-25

**Decision:**

Accept (poster)

**Comment:**

This paper proposes a self-training approach to enhance the image comprehension abilities of VLMs by prompting the model with good and bad prompts to generate a preference datasets. The good prompts guide the model to reason step-by-step while the bad prompts are intentionally misleading or used with corrupted images. In addition, the image descriptions of good prompts are used to augment existing instruction tuning data. Experiments show the effectiveness of STIC across many established VLM benchmarks despite using a small amount of additional unlabeled data.

Many reviewers appreciate the novelty of the STIC approach to construct preference dataset (qFQt), its data efficiency using unlabeled images (s8oQ, qFQt, B7oX) and finally the significant performance gains (B7oX, s8oQ) on common benchmarks. The reviewers are happy with the authors' rebuttal and most of them are leaning to accept the paper with overall scores (6, 6, 6, 4). The WR reviewer's main concern was about the instruction-following ability of the model when good prompts are given. The author had provided a detailed response in their rebuttal and global response A1, which has addressed the concern adequately in my opinion.

Thus, I recommend to accept this paper.